

# Dynamics of phytoplankton and heterotrophic bacterioplankton in the western tropical South Pacific Ocean along a gradient of diversity and activity of diazotrophs

France Van Wambeke [1], Audrey Gimenez[1], Solange Duhamel[2], Cécile Dupouy[1,3], Dominique Lefevre [1], Mireille Pujo-Pay[4], Thierry Moutin [1]

[1] Aix-Marseille Université, CNRS/INSU, Université de Toulon, IRD, Mediterranean Institute of Oceanography (MIO) UM 110, 13288, Marseille, France
[2] Lamont-Doherty Earth Observatory, Division of Biology and Paleo Environment, Columbia University, PO Box 1000, 61 Route 9W, Palisades, New York 10964, USA
[3] Aix Marseille Université, CNRS/INSU, Université de Toulon, IRD, Mediterranean Institute of Oceanography (MIO) UM 110, 98848, Nouméa, New Caledonia
[4] CNRS, Laboratoire d'Océanographie Microbienne (LOMIC), Sorbonne Universités, UPMC Univ Paris 6,
Observatoire Océanologique, 66650, Banyuls/mer

*Correspondance to*: France Van Wambeke (france.van-wambeke@mio.osupytheas.fr)

**Abstract.** Heterotrophic prokaryotic production (BP) was studied in the Western Tropical South Pacific using
the leucine technique. Integrated over the euphotic zone, BP ranged from 58–120 mg C m$^{-2}$ d$^{-1}$ within the Melanesian Archipelago, and from 31–50 mg C m$^{-2}$ d$^{-1}$ within the subtropical gyre. Nitrogen was often one of the main factor controlling BP on short time scale as shown using enrichment experiments, followed by dissolved inorganic phosphate (DIP) near the surface and labile organic carbon deeper in the euphotic zone. With N$_2$ fixation being one of the most important fluxes fueling new production, we explored relationships between BP,
primary production (PP) and N$_2$ fixation rates. BP variability was better explained by the variability of N$_2$ fixation rates than by that of PP in surface waters of the Melanesian Archipelago, which were characterized by N depleted layers, and low DIP turnover times (T$_{DIP}$ < 100 h). However, BP was more significantly correlated with PP but not with N$_2$ fixation rates where DIP was more available (T$_{DIP}$ > 100 h), i.e. in a layer deeper than the euphotic zone -including the deep chlorophyll maximum depths- in the Melanesian Archipelago, or within the
entire euphotic zone in the subtropical gyre. Bacterial growth efficiency (BGE) ranged from 6 −10 %. Applying correcting factors to estimate gross primary production and correcting BP for *Prochlorococcus* assimilation of leucine, we showed a large variability in the contribution of gross primary production to bacterial C demand. Exploration of a bloom collapse at one site south of Vanuatu showed the importance of blooms, which can persist over extensive distance for long periods of time, and can maintaining net autotrophy where they occur.
Using a Lagragian sampling strategy during 6 days, long duration sites allowed for the study of the rapid changes including BP, primary production and BGE, that occurred during the bloom collapse.

## 1 Introduction

Heterotrophic prokaryotes can process, on average, 50 % of the carbon (C) fixed by photosynthesis in many aquatic systems (Cole, 1988). Understanding the controls of heterotrophic bacterial growth, their
production, and their respiration rates is fundamental for two major aspects of marine C cycling: i) to explore one of the possible fate of primary production through the microbial food web, and ii) to construct a metabolic balance based on C fluxes. To asses these two major features, bacterial carbon demand (BCD, i.e. the sum of



heterotrophic bacterial production (BP) and bacterial respiration (BR)) is compared to primary production (PP). The metabolic state of the ocean, and in particular the status of net heterotrophy within oligotrophic systems, has

been largely debated in the last decade (see for example review in Duarte et al., 2013; Ducklow and Doney, 2013; Williams et al., 2013).

The South Pacific gyre (GY) is ultra-oligotrophic, and is characterized by deep UV penetration, by deep chlorophyll maximum depth (dcm) down to 200 m, and by 0.1 µM nitrate (NO3) isocline at 160 m (Claustre et al., 2008b; Halm et al., 2012). Our knowledge of the south Pacific Ocean's metabolic state based on C fluxes is

fragmentary, since only little primary production data has previously been reported, and never simultaneously with BP (see references in Table 1). The exception is the BIOSOPE cruise conducted in the GY and eastern tropical south Pacific (ETSP) in Nov./Dec. 2004, where both PP and BP have been estimated simultaneously (Van Wambeke et al., 2008b).

The waters coming from the GY are essentially transported by the South Equatorial current toward the

Melanesian archipelagos in the Western Tropical South Pacific (WTSP). Interest in this region has increased due to field and satellite observation showing intermittent phytoplankton blooms in the area (Dupouy et al., 2008; 2011; Tenorio et al., 2018). The WTSP is a highly dynamic region (Rousselet et al., 2017) where patches of chlorophyll blooms can persist for up to a few weeks (de Verneil et al., 2017b). The WTSP has recently been shown to be a hotspot for biological nitrogen fixation ($N_2$fix, Bonnet et al., 2017), extending to this whole

oceanic region what was already shown locally near New Caledonia (Garcia et al., 2007). Based on nitrogen budgets, such blooms can sustain significant, new production and export in this area (Caffin et al., 2017). The development of these blooms are explained by different hypotheses, including temperature thresholds (in particular regulating *Trichodesmium* blooms); increased light providing more energy; the stratification of surface waters favoring depletion of nitrate and reducing competition with non-fixing primary producers; and increased

availability of Fe and P due to island mass effects, volcanic activities or atmospheric nutrient deposition (Moutin et al., 2005; 2008; Luo et al., 2014; Martino et al., 2014; Shiozaki et al., 2014, Bonnet et al., this issue).

While the dynamics of heterotrophic prokaryotes coupling with primary producers has been explored in many regions of the ocean, these processes have not been studied in the WTSP. Because most oligotrophic oceans are nitrogen limited, PP and $N_2$fix have already been sampled simultaneously in diverse studies and their

relationships examined. Taking a Redfield ratio of 6.6, the contribution of $N_2$fix rates to PP, integrated over the euphotic zone, has been found to range from 1−9 % in diverse provinces of the Atlantic (Fonsesca et al., 2016). The ratio is from 15− 21 % in the WTSP and from 3–4 % in the centre of the GY (Raimbault and Garcia, 2008; Caffin et al., 2017). Few studies have attempted to examine how the variability of nitrogen fixation can be linked to that of heterotrophic activity, or to identify the contribution of $N_2$fix rates to heterotrophic prokaryotic N

demand. Yet, recent genomic analyses exploring the diversity of the nitrogenase reductase (*nifH*) gene have revealed the importance of non-cyanobacterial nitrogen fixers (Gradoville et al., 2017 and ref therein). Owing to the fact that a great abundance of *nifH* gene copies does not imply that $N_2$ fixation is occurring (see for example Turk-Kubo et al., 2014), diverse tests have been conducted to assess heterotrophic $N_2$ fixation indirectly. For example, in the oligotrophic Eastern Mediterranean Sea, aphotic $N_2$ fixation can account for 37 to 75 % of the

total daily integrated $N_2$fix rates (Rahav et al., 2013). In the Red Sea, $N_2$fix rates are correlated to BP but not to PP during the stratified summer season, while during a *Trichodesmium* bloom in winter, both PP and BP increased with $N_2$fix rates although the correlation was still insignificant with PP (Rahav et al., 2015). In the





South Pacific, the presence of non-cyanobacterial nitrogen fixers has been detected in the dark ocean as the in euphotic layer, with detectable levels of *nifH* gene expression, as measured by qPCR or N$_2$fix activity

determined in darkness (in the GY: Halm et al., 2012, Moisander et al., 2014; in the Eastern Tropical South Pacific Ocean: Bonnet et al., 2013, in Bismarck and Solomon Seas: Benavides et al., 2015). The addition of selected organic molecules such as glucose (Dekaezemacker et al., 2013) or natural organic matter such as Transparent Expolymer Particles, can influence N$_2$fix rates (Benavides et al., 2015). Finally, recent experiments based on incubation with $^{15}$N-labeled N$_2$ coupled to nano-SIMS analyses also demonstrated that a rapid transfer,

at the scale of 24 to 48 h, can occur between N$_2$ fixers, non-fixing phytoplankton and heterotrophic prokaryotes (Bonnet et al., 2016).

       In this study, we examined the horizontal and vertical distribution of heterotrophic prokaryotic production alongside photosynthetic rates, N$_2$fix rates and phosphate turnover times across the WTSP, in order to relate these fluxes with bottom-up controls (related to nitrogen, phosphate and labile C availability) Particular

attention was given to determine the coupling between BP and PP or nitrogen fixation rates, to examine the variability of bacterial carbon demand in comparison to gross primary production ratios, and finally to discuss the metabolic state of this region

## 2 Materials and methods


### 2.1 Sampling strategy

       The OUTPACE cruise (doi.org/10.17600/15000900) was conducted in the WTSP region, from February 18$^{th}$ to April 3$^{rd}$, 2015, along a transect extending from the North of New Caledonia to the western part of the South Pacific Gyre (WGY) (25°S 115 E – 15°S, 149°W, Fig. 1). For details on the strategy of the cruise,

see Moutin et al. (2017). Stations of short duration (< 8 h, 15 stations named SD1 to SD15, Fig. 1) and long duration (6 days, 3 stations named LDA to LDC) were sampled. Generally, at least 3 CTD casts going down to 200 m were conducted at each short station, except at SD5 and SD9 (two casts) and at SD13 (one cast). The LD stations were abbreviated as LDA (north New Caledonia), LDB (Vanuatu area), LDC (oligotrophic reference in the WGY area) and were chosen along a regional gradient in oligotrophy. LD stations were selected using

satellite imagery, altimetry and Lagragian diagnostics (Moutin et al; 2017), as well as on the abundance of selected diazotrophs *nifH* gene copies, analyzed by quantitative Polymerase Chain Reaction (qPCR), in real time on board (Stenegren et al. 2017). At these LD stations, CTD casts were performed every 3 hours during at least 5 days. All samples were collected from a CTD-rosette system fitted with 20 12-L Niskin bottles and a Sea-Bird SBE9 CTD.

At the SD stations, water samples used for measuring *in situ*–simulated primary production (PP$_{deck}$), dissolved inorganic phosphate turnover times (the ratio of DIP concentration to DIP uptake rate, T$_{DIP}$), N$_2$fix rates came from the same rosette cast as used for measuring BP. On top of those experiments, at the LD sites, we also conducted biodegradation experiments to determine bacterial growth efficiency (BGE), as well as enrichment experiment to explore the factors limiting BP.

Besides measurements of chlorophyll *a,* BP, PP, T$_{DIP}$ and DOC described below, other data presented in this paper include hydrographic properties, nutrients, N$_2$fix, for which detailed protocols of analysis are available





in Moutin et al. (2017; this issue) and Bonnet et al. (this issue). In the discussion section, we used dark community respiration rates which detailed protocols are available in Lefevre et al. (this issue).

**2.2 Chlorophyll a**

For chlorophyll a (chl a), a sample of 288 mL of seawater was filtered through 25 mm Whatman GF/F filters immediately after sampling and placed at -80°C in Nunc tubes until analysis. At the laboratory (3 months after the cruise), after grinding the GF/F filter in 5 ml methanol, pigments (chl a and phaeophytin) were extracted in darkness over a 2 h period at 4°C and analyzed with a Trilogy Turner 7200-000 fluorometer according to Le

Bouteiller et al. (1992). Sampling for chl a analysis started only at station LDA (Dupouy et al., this issue).

Due to the heterogeneity at the time of sampling and the nature of the populations present, i.e. essentially different fluorescence yields over depth and species (Neveux et al., 2010), the overall correlation of in vivo fluorescence (chl iv) with chl a was very patchy (chl a=1.582 * chl iv + 0.0241, n = 169, r = 0.61).

In addition, restricted to a given site, the conversion factor relative to these values was satisfactory for the whole euphotic layer at LDA and LD C, but not LDB

Site LDC chl a = 0.809 * chl iv + 0.0238, n = 60, r = 0.91

Site LDB chl a = 2.267 * chl iv + 0.1039, n = 32, r = 0.51

Site LDA chl a = 0.947 * chl iv + 0.1292, n = 25, r = 0.91

Thus in vivo fluorescence was used only to track high frequency variability at the LD sites, the shape of vertical profile's distributions and the location of the dcm, as well as longitudinal trends. Fluorometric discrete data (chl a) was always used in correlations between chlorophyll biomass and other variables, or when calculating integrated stocks.

**2.3 Bacterial production**

Bacterial production (BP, *sensus stricto* referring to heterotrophic prokaryotic production) was determined onboard using the microcentrifuge method with the $^3$H- leucine ($^3$H-Leu) incorporation technique to measure protein production (Smith and Azam, 1992). Triplicate 1.5 mL samples and a killed control with trichloracetic acid (TCA) at 5 % final concentration were incubated with a mixture of [4,5-$^3$H]leucine

(Amersham, specific activity 112 Ci mmol$^{-1}$) and nonradioactive leucine at final concentrations of 7 and 13 nM, respectively. Samples were incubated in the dark at the respective *in situ* temperatures for 1–4 h. Occasionally, we checked that the incorporation of leucine was linear with time. Incubations were ended by the addition of TCA to a final concentration of 5 %, followed by three runs of centrifugation at 16000 g for 10 minutes. Bovine serum albumin (BSA, Sigma, 100 mg l$^{-1}$ final concentration) was added before the first centrifugation. After

discarding the supernatant, 1.5 ml of 5 % TCA was added before the second centrifugation, and for the last run, after discarding the supernatant, 1.5 ml of 80 % ethanol was added. The ethanol supernatant was discarded and 1.5 ml of liquid scintillation cocktail (Packard Ultimagold MV) was added. The radioactivity incorporated into macromolecules was counted in a Packard LS 1600 Liquid Scintillation Counter on board the ship. A factor of 1.5 kg C mol leucine$^{-1}$ was used to convert the incorporation of leucine to carbon equivalents, assuming no

isotopic dilution (Kirchman, 1993). Indeed, isotopic dilution ranged from 1.04 to 1.18 as determined occasionally with preliminary experiments checking the saturating level of $^3$H-leucine. Standard errors





associated with the variability between triplicate measurements averaged 13 % and 6 % for BP values lower and higher than 10 ng C l$^{-1}$ h$^{-1}$, respectively. At the LD sites, BP was sampled every day at 12:00 local time.

**2.4 Primary production ant phosphate turnover times**

Primary production (PP) and dissolved inorganic phosphate turnover times ($T_{DIP}$) were determined using a dual $^{14}$C–$^{33}$P labelling technique following Duhamel et al. (2006) and described in Moutin et al. (this issue). Briefly, after inoculation with 10 μCi of $^{14}$C sodium bicarbonate and 4 μCi of $^{33}$P-orthophosphoric acid, 150-mL polycarbonate bottles were incubated in on-deck incubators equipped with blue screens (75, 54, 36, 19, 10, 2.7, 1, 0.3 and 0.1 % incident light, https://outpace.mio.univ-amu.fr/spip.php?article135) and flushed continuously with surface sea water. Samples were then filtered through 0.2 μm polycarbonate membranes, with radioactivity retained by the filters being assessed by liquid scintillation counting directly on board and after 12 months in the laboratory. Rates of daily primary production were computed using the conversion factors $\tau_{(Ti\,;T)}$ according to Moutin et al. (1999) to calculate normalized (dawn-to-dawn) daily rates from the hourly rates measured in the on-deck incubators ($PP_{deck}$). Measurements of PP using the JGOFs protocol (*in situ* moored lines immerged for 24 h from dusk to dusk, $IPP_{in\,situ}$) were also performed at each long stations on days 1, 3 and 5 (see Caffin et al., 2017 for details). Integrated rates within the euphotic zone were estimated by trapezoidal integrations, assuming the same rate between 0 m and the shallowest layer sampled, and considering PP to be zero at 20 m below the deepest layer sampled.

**2.5 Bacterial growth efficiency**

Bacterial growth efficiency (BGE) and DOC lability were estimated at the three LD sites using dilution experiments with seawater sampled in the mixed layer. The seawater used for the experiments was sampled from Niskin bottles (9 m at LDA, 7 m at LDB and 16 m at LDC) from a CTD cast done at 12:00 local time on the first day of measurements at each LD site (CTD cast numbers 27, 109 and 158, respectively). Using the same seawater sample, a bacterial inoculum (400 ml of a < 0.8 μm filtrate) was mixed with 2.6 L of < 0.2 μm filtrate, in a borosilicate bottle. We incubated them in the dark, for up to 10 days, in a laboratory incubator set at *in situ* temperature. Periodically, for up to 10 days, subsamples were taken to estimate DOC concentrations and bacterial production. The BGE was estimated from DOC and bacterial production estimates on a given time interval corresponding to the exponential phase of BP as follows:

$$BGE = BP_{int} / DOC_{cons} \qquad\qquad (eq\ 1)$$

where $BP_{int}$ is the trapezoidal integration of BP with time for the period considered, and $DOC_{cons}$ the dissolved organic carbon consumed during that period, corresponding to the difference in DOC concentration between initial ($DOC_{initial}$) and minimal DOC ($DOC_{min}$). From these experiments we determined also the labile fraction of DOC was determined as:

$$(DOC_{initial} - DOC_{min}) / DOC_{initial} \qquad\qquad (eq\ 2)$$

Samples for dissolved organic carbon were filtered through two precombusted (24h, 450°C) glass fiber filters (Whatman GF/F, 25 mm) using a custom-made all-glass/Teflon filtration syringe system. Samples were collected into precombusted glass ampoules and acidified to pH 2 with phosphoric acid ($H_3PO_4$). Ampoules were immediately sealed until analyses by high temperature catalytic oxidation (HTCO) on a Shimadzu TOC-L analyzer (Cauwet, 1999). Typical analytical precision is ± 0.1–0.5 (SD) or 0.2–0.5 % (CV). Consensus reference





materials (http://www.rsmas.miami.edu/groups/biogeochem/CRM.html) was injected every 12 to 17 samples to
insure stable operating conditions.

205 **2.6 Enrichment experiment**

Enrichments experiments were performed along vertical profiles at the three LD sites LDA, LDB and
LDC. Seawater was sampled at 12:00 local time on day 2 of measurements at each site (CTD casts numbers 33,
117 and 166, respectively). Nutrients were added in 60 ml transparent polycarbonate bottles at a final
concentration of 1 $\mu$M $NH_4Cl$ + 1$\mu$M $NaNO_3$ in 'N' amended bottles, 0.25 $\mu$M $Na_2HPO_4$ in 'P' amended bottles,
210 10 $\mu$M C-glucose in 'G' amended bottles. The sum of all these elements were added in 'NPG' amended bottles.
Controls 'C' were left unamended. Bottles were incubated on average for 24 h under simulated *in situ* conditions
(in the same on-deck incubators than those used for $PP_{deck}$). Selected depths chosen encompassed the euphotic
zone. At LDA: 9, 24, 35, 70, 100 m were incubated under 54, 10, 3 1 and 0.3 % incident light; at LDB: 7, 12, 27,
42 m were incubated under 54, 36, 10, and 3 % incident light, and at LDC: 16, 60, 91 and 135 m were incubated
215 under 54, 10, 3, and 1 % incident light, respectively. For depths deeper than the euphotic zone (200 m at LDA,
100 m and 200 m at LDB, and 200 m at LDC), flasks were incubated in the dark in a laboratory incubator set at
*in situ* temperature. After 24 h of incubation, subsamples were taken from each flask to perform BP incubations
as described for *in situ* samples (triplicate estimates, incubation in the dark), except that incubations lasted only 1
h. Results are presented as enrichment factor relative to the unamended control.


**2.7. Statistics**

Relationships between variables were established using model II Tessier linear regressions, from log-
transformed data. Multiple regressions were also used to study the simultaneous effects of PP and $N_2$fix rates on
BP variability. The effect of enrichments was tested comparing BP obtained in the unamended control with BP
obtained in the amended samples using a Mann-Whitney non parametric test.

**3. Results**

**3.1 Regional oceanographic settings**

The transect encompassed a longitudinal gradient starting near New Caledonia, crossing the Vanuatu
and Fidji Arcs and finishing inside the west part of the ultra-oligotrophic South Pacific Gyre (Fig. 1).
Temperatures ranged 19.7–30.2°C within the 0–200 m layer (see Fig 3a in Moutin et al., 2017). Density revealed
shallow mixed layers, due to the sharp temperature gradients. The mixed layers for most of the cruise were $\leq$ 20
m, Moutin et al., this issue, de Verneil et al., 2017a), excepted at SD13 and LDC: 27 and 34 m respectively. The
dcm varied from 10 to 154 m (Table 2), with minimum values at LDB and deeper values in the WGY area
(Table 2, see Fig 3d in Moutin et al., 2017).

The transect covered a vast region of the WTSP and the main transition in distribution of
biogeochemical and biological properties corresponded to sub regions geographically separated by the Tonga
volcanic arc: the Melanesian Archipelago (MA) area was covered by stations SD1 to SD12, and included LDA.
A detailed analysis of the vertical distribution of nutrients and organic matter (Moutin et al., this issue) made it
possible to separate two types of stations within the MA area: one group between 160 and 170°E called WMA





for 'Western Melanesian Archipelago' clustered SD1, 2, 3 and LDA and a second group South of Fidji called EMA for 'Eastern Melanesian Archipelago' clustered SD6, 7, 9 and 10. The western part of the south Pacific Gyre (WGY) was covered by stations SD13 to SD15 and included LDC. LDB, although included geographically within MA area, corresponded to a particular bloom condition and is therefore presented and discussed separately.

### 3.2 Longitudinal distributions

Averaged per SD station, the dcm fluctuated between 61 and 115 m in the MA area and between 123 and 154 m in the WGY area. The integrated chlorophyll a concentrations (determined by fluorimetry) ranged from 13–23 mg chl a m$^{-2}$ in the WGY area, and were significantly lower than in the MA area (20–38 mg chl a m$^{-2}$, Mann-Whitney test, p = 0.013). The mean dcm within EMA was slightly deeper (105 ± 10 m, Table 2) than in WMA (82 ± 10 m, Mann-Whitney test, p = 0.03)

Primary production rates ranged from undetectable to 20.8 mg C m$^{-3}$ d$^{-1}$ (Fig 2a). PP rates higher than 10 mg C m$^{-3}$ d$^{-1}$ were obtained in the MA area at SD1, SD7, and SD9 within the surface or sub surface (but also at sites LDA and LDB, see below), whereas stations SD13 and eastwards presented values lower than 1.3 mg C m$^{-3}$ d$^{-1}$. Bacterial production ranged 0.8–138 ng C l$^{-1}$ h$^{-1}$ in the 0–200 m layer (Fig. 2b). Within the MA area, BP reached values higher than 100 ng C l$^{-1}$ h$^{-1}$ at SD1 and SD5 within the surface (5 m depth, Fig. 2b). High BP values were also obtained at LDB (see below). Within the WGY, maximum BP rates reached 27 ng C l$^{-1}$ h$^{-1}$ (at site LDC, see below).

Integrated primary production (IPP$_{deck}$) ranged from 178–853 mg C m$^{-2}$ d$^{-1}$ within the MA area and from 104–213 mg C m$^{-2}$ d$^{-1}$ within the WGY area (Fig. 3a). Integrated BP (IBP) over the euphotic zone ranged from 58–120 mg C m$^{-2}$ d$^{-1}$ within the MA area and from 31–35 mg C m$^{-2}$ d$^{-1}$ within the WGY area (Fig. 3a). Both integrated fluxes within the euphotic zone were statistically lower within the WGY (Mann-Whitney test, p = 0.01 for IBP, and p = 0.03 for IPP$_{deck}$). In contrast, for the WMA and EMA group of stations, integrated fluxes were not statistically different, neither for IBP (99 ± 15 versus 95 ± 12 mg C m$^{-2}$ d$^{-1}$, Mann-Whitney test, p > 0.05) nor for IPP$_{deck}$ (481 ± 47 versus 471 ± 276 mg C m$^{-2}$ d$^{-1}$, p > 0.05)

DIP turnover times (T$_{DIP}$) ranged over 4 orders of magnitude along the transect (from 2.1 up to 1000 h, Fig. 4). Vertical profiles roughly increased with depth, coincident with the increase of DIP concentrations below the phosphacline. T$_{DIP}$ also showed a clear MA–WGY transition zone. Within the WGY mixed layers, T$_{DIP}$ ranged from 469–4200 h, coincident with detectable amounts of DIP (around 100 nM) in this area (Moutin et al., this issue). Within MA, T$_{DIP}$ were lower than in the WGY area. However, T$_{DIP}$ ranged from 2–857 h in the mixed layers of the MA area, with lower values associated to stations LDB-d5, LDA-d5, SD3, SD4 and SD6, and higher values associated to SD2 SD5, SD7 and SD12. This T$_{DIP}$ range encompassing two orders of magnitude suggested a much higher range of DIP demand, than DIP concentration alone would suggest.

### 3.3 Daily variability at the long occupation sites.

Site LDA presented variable dcm with patches of in vivo fluorescence going up and down in the water comumn with time along a band of 40 m height (dcm varied between 63 and 101 m, Table 2), However, the dcm corresponded to a stable density horizon (σt 23.55 ± 0.04 kg m$^{-3}$), and thus this fluctuation in dcm corresponded to internal waves cgaracterized by a periodicity of about 2 per day (Fig. 5). The fluorescence intensities



increased during each afternoon, showing a diel cycle in net chlorophyll production, but no long-term trend was observed during the 5 days. However, BP and PP peaked in shallower layers, at 10–25 m depth (range 47–68 ng C l$^{-1}$ h$^{-1}$ for BP, and 3–6 mg C m$^{-3}$ d$^{-1}$ for PP) and sometimes presented a second, much less intense peak, around

the dcm (Fig. 5). Overall, BP and PP showed parallel trends, increasing slightly on day 3 compared to days 1 and 5. Integrated chl a was on average $26.0 \pm 2.6$ mg chl a m$^{-2}$, IPP$_{in\ situ}$ $267 \pm 79$ mg C m$^{-2}$ d$^{-1}$ (note, however the high value obtained on day 5 with deck incubation (IPP$_{deck}$ 698 mg C m$^{-2}$ d$^{-1}$) and integrated BP $98 \pm 16$ mg C m$^{-2}$ d$^{-1}$ (Table 2).

Site LDB was sampled inside a high chlorophyll patch. It presented maxima of in vivo fluorescence between 10 and 77 m, and this depth showed a significant linear deepening with time ($10.4 \pm 0.8$ m d$^{-1}$, r = 0.89,

n = 45, p < 0.001). Contrarily to the site LDA, this dcm did not correspond to a stable pycnocline horizon, as density associated with the dcm varied between 21.8 and 23.9 kg m$^{-3}$, reaching a plateau after day 4 (data not shown). Chlorophyll was distributed over a larger layer (between the surface and 80 m) during the first three days, and then presented a narrow and deeper zone of accumulation, with intensities increasing due to photo-acclimation. Similarly to site LDA, in vivo fluorescence intensity peaks increased slightly during the afternoons

(Fig. 6). Integrated chl a decreased from 53.2 to 23.9 mg chl a m$^{-2}$ between days 1 and 5, which corresponded to a chlorophyll biomass loss of about 7.3 mg chl m$^{-2}$ d$^{-1}$. The shape of BP and PP vertical profiles was particularly modified at day 5, showing a small decrease of subsurface values for BP (125 down to 100 ng C l$^{-1}$ h$^{-1}$) but a larger one for PP (15 down to 9 mg C m$^{-3}$ d$^{-1}$). In contrast, BP increased within the dcm at day 5. Integrated PP

decreased by approximatively 145 mg C m$^{-2}$ d$^{-1}$ between days 3 and 5. At day 4, PP was not measured but a decrease of BP rates in sub-surface layers was already visible. Six profiles were available for BP from which we estimated a linear increasing trend of 7.2 mg C m$^{-2}$ d$^{-1}$ per day (n = 6, r = 0.78).

Site LDC, typical of the WGY area, presented a deeper dcm, ranging between 115 and 154 m, due to internal waves, and like for site LDA its density was very stable (σt $24.59 \pm 0.02$ kg m$^{-3}$, n = 46). At the dcm, in

vivo fluorescence showed a diel rhythm, increasing during the afternoons and decreasing during the nights (Fig. 7). PP exhibited two peaks around 40–60 m and 120 m, but remained very low (max 2.3 mg C m$^{-3}$ d$^{-1}$). BP profiles paralleled those of PP, reaching also small maxima at 60 m and occasionally a second one at 120 m. Maximum BP rate was 27.7 ng C l$^{-1}$ h$^{-1}$. IPP$_{in\ situ}$ ranged 149–165 mg C m$^{-2}$ d$^{-1}$. Integrated BP values were also very low ($44 \pm 5$ mg C m$^{-2}$ d$^{-1}$) and both integrated rates exhibited no trend with time.


### 3.4 Relationships between BP, PP, N$_2$fix and T$_{DIP}$

Log-log relationships between BP and PP presented a continuous linear trend for all samples with values below or above a T$_{DIP}$ of 100 h (Fig. 8a). Values below 50–100 h are representative of a restricted access to DIP by microorganisms (Moutin et al., 2008). A T$_{DIP}$ below 2 days was shown to be critical for

*Trichodesmium* spp. growth (Moutin et al., 2005). The depth at which this threshold was reached varied from surface to 64 m in MA although all T$_{DIP}$ values were higher than 100 h in the SPG. Relationships were:

$$\log BP = 0.835 \log PP - 0.55,\ n=64,\ r = 0.37\ \text{and}$$

$$\log BP = 0.829 \log PP - 0.52,\ n=122,\ r = 0.70$$

for samples where T$_{DIP}$ was ≤ 100 h and > 100 h, respectively. In contrast, log-log relationships linking BP and

N$_2$fix presented different trends for samples corresponding to depths where T$_{DIP}$ was below or above 100 h. Relationships were:




$$\log BP = 0.752 \log N_2 fix - 0.78, \; n=39, \; r = 0.52 \text{ and}$$

$$\log BP = 0.438 \log N_2 fix - 0.31, \; n=55, \; r = 0.43$$

for samples where $T_{DIP}$ was $\leq 100$ h and $> 100$ h, respectively (Fig. 8b). This suggests that BP was more

dependent on $N_2$fix than on PP in the surface, P-depleted waters. As PP and $N_2$fix could co-vary, a multiple

regression was tested (Table 3). The partial coefficient was not significant for $N_2$fix for samples with $T_{DIP} > 100$

h. The partial coefficients were both significant for $N_2$fix and for BP for samples characterized by $T_{DIP} \leq 100$ h,

but $N_2$fix better explained the distribution of BP in the multiple regression analysis compared to PP (t-test,

p=0.024 for PP and p < 0.0001 for $N_2$fix).

Integrated $N_2$fix accounted for 3.3 to 81 % of the bacterial nitrogen demand, assuming a stoichiometric

molar C/N ratio of 5 for heterotrophic prokaryotic biomass (Fig. 3b). This large longitudinal variability was also

temporal, as showed by the variability among the three LD sites: the ranges were from 28–46 % at LDA, and

from 6–11 % at LDC, with no particular temporal trend, but a decrease was clearly observed at LDB, from 68 to

19 % between day 1 and 5.

We also examined relationships linking $T_{DIP}$ with other biological fluxes using a multiple regression

[$\log T_{DIP} = f(\log PP, \log N_2 fix, \log BP)$], incorporating 91 samples where the three rates were measured

simultaneously (Table 4). The partial coefficients were both significant for both $N_2$fix (p < 0.0001) and for BP (p

= 0.003) but not for PP (p = 0.23). As all biological rates decreased with depth, we also examined this correlation

using data within the mixed layer to avoid the depth effect. With this restricted data set (47 samples) the partial

coefficients were significant only for BP (p = 0.0024) and just under the significance threshold for $N_2$fix (p =

0.056) and still insignificant for PP.

### 3.5 DOC lability and BGE

In the three biodegradation experiments starting on day 1 at each LD site using sub-surface waters, BP

and abundances increased significantly, and growth rates (determined from exponential phase of BP increase)

ranged 0.08 – 014 $h^{-1}$. DOC was slightly consumed. A decrease of only 2 to 5 % in DOC concentrations was

measured in over 10 days, with this labile fraction percentage being the lower in site C (Table 5). Bacterial

growth efficiencies were 13, 6.3 and 6.7 % in sites LDA, LDB and LDC, respectively (Table 5).

### 3.6 Enrichment experiments

Vertical distributions of in vivo fluorescence, nutrients and BP sampled from a CTD cast starting at

12:00 pm on day 2 of each LD site, captured conditions prevailing before enrichments, and are presented

together with enrichment factors obtained at the different depths tested (Fig. 9). As DOC was not sampled on

this profile, DOC data for the whole site are presented instead. DOC peaked near the surface at site LDA (77 ± 1

μM), was more variable but higher near the surface at site LDB, with maximum values covering a larger surface

layer, from surface to 27 m, with average DOC of 84 ± 1 μM. Site LDC presented a large sub surface maximum

within 28–42 m, reaching 77 ± 2 μM. Nitrate concentrations were below the detection limits of standard methods

in upper layers. Vertical distributions exhibited a nitracline occurring at different depths and coincident with the

dcm at LDA and LDC (100 m at LDA, 135 m at station LDC) but not at LDB (a large peak of chlorophyll within

20–70 m and a nitracline at 100 m). Slight peaks of nitrite also occurred in the vicinity of the nitracline.

Phosphate concentrations exhibited more contrasted vertical profiles than nitrate: DIP was higher than 100 nM in



the surface layers of LDC, and presented a phosphacline shallower than the nitracline at LDA and LDB, with
DIP reaching concentrations below the analytical detection limits in the mixed layer (i.e. < 50 nM, see Moutin et
al., this issue for more details on nutrient distribution).

365       At site LDA, nitrogen alone was the first factor stimulating BP down to 100 m depth, which
corresponded to the dcm and a nitrite regeneration layer. Although significant at 9, 24, 35 and 100 m depth
(Mann-Whitney test, $p < 0.05$) the response to N amendments was small, at best an enrichment factor of x1.6.
Glucose alone stimulated BP at 35 m but only by a factor x1.4. However, below the euphotic zone, glucose was
the first factor stimulating BP (enrichment factor x2.4 at 200 m). NPG showed the largest enrichment factors,
x2.5 to x3.0 and all along the profile except at 9 m were NPG amendments did not significantly affected BP
compared to the control.

      At site LDB, between the surface and 42 m, nitrogen alone but also phosphate alone stimulated BP to a
larger extent than at site LDA, but only by a factor x1.5–3 for P, and x1.8–3.7 for N. At 100 m and 200 m,
nitrogen continued to stimulate BP to a small extent (x3.0, and x2.2, respectively), but the maximum
enhancement was obtained after glucose addition alone (x59, and x107, respectively). At these depths, the BP
response after addition of NPG was also largely amplified compared to shallower layers (x120–132 compared to
x3.7–6.8, respectively).

      At site LDC, BP reacted mostly to glucose alone, with enhancement factors increasing from x2.6 at 16
m to x24 at 200 m. Nitrogen alone also stimulated BP, but to a smaller extent than glucose, even within surface
layers (x1.2 to x9). In comparison to single amendments, the NPG addition particularly stimulated BP at 60 m
and 90 m depth.

### 4   Discussion

**4.1 An overview of BP and PP fluxes in the WTSP**

      Here, we provide a unique, coherent dataset with simultaneous estimates of PP, BP, $T_{DIP}$ and $N_2$fix rates
in the WTSP. Indeed, recent interest in describing fluxes and planktonic communities responsible for $N_2$fix rates
in diverse environments has increased, particularly in oligotrophic open oceans, although measurements in the
tropical area of the South Pacific Ocean are rare (summarized in Gruber et al., 2016 and more recently Bonnet et
al., 2017). To date, however few studies have attempted to simultaneously study the consequences of such
activities on the functioning of the microbial food webs.

      We distinguished at least two main areas. The ultra-oligotrophic area, (stations SD13 to SD15 and site
LDC, i.e. the eastern part of the transect), was characterized by dcm deeper than 115 m, deep nitracline (130 m)
and nitrite peaks around 150 m and detectable amounts of phosphate at the surface (> 100 nM, Moutin et al., this
issue). On the other hand, stations in the western part of the transect displayed higher fluxes of PP, BP and $N_2$fix
rates in general, but were not disposed along a clear trophic gradient, although we could identify two sub-areas
corresponding to EMA and WMA areas, EMA-type stations having intermediary dcm depths (between WMA
and WGY), and distinct nutrient distributions in addition to intermediary nitracline depths (Moutin et al., this
issue). This variability in longitudinal distribution, as seen from the nitracline and the dcm depths, was also



reflected by $T_{DIP}$ variability ranging two orders of magnitude in the top-40 m layers within MA (2 to 700 h). The role of the submesoscale activity largely explained such variability (Rousselet et al., 2017).

Previous *in situ* measurements of primary production in the tropical south Pacific, not directly focusing on coastal areas or within upwelling areas in the East, are rare, and summarized in Table 1. These daily
particulate primary production rates, based on the $^{14}C$ or $^{13}C$ technique, confirm the trend that we observed in the WTSP, i.e. extremely low values in the central GY area ranging 8–167 mg C $m^{-2}$ $d^{-1}$ (Table 1). Around it, PP increased, but still under oligo to mesotrophic conditions, in the Eastern region of the GY, in the South of the GY, and in the western part of the WTSP around New Caledonia and between New Caledonia and Australia (Table 1). Further northwest, in the Solomon Sea, PP increased to much higher values in an area of intense
nitrogen fixation, up to 3000 mg C $m^{-2}$ $d^{-1}$ (Table 1). However, although an increasing number of PP and $N_2$fix values are available in the WTSP and within the GY, to our knowledge, the only other BP data available in these regions are those estimated during the BIOSOPE cruise (Nov./Dec. 2004), along a longitudinal transect further east between Tahiti and Chile (Van Wambeke et al., 2008b). In this study BP integrated across the euphotic zone ranged from 86–144 mg C $m^{-2}$ $d^{-1}$ within the Marquesas Archipelago area, from 43–114 mg C $m^{-2}$ $d^{-1}$ within the
center of the GY and 57–93 within the eastern part of the GY. Therefore, in the WTSP we encountered the same range of BP values than in the GY area eastern of 140°W.

### 4.2 BGE and Metabolic state

Bacterial growth efficiencies (BGE) obtained from biodegradation experiments were low, ranging 6–12 %, with a small labile fraction of DOC (only 2–5 % of biodegradable DOC in 10 days). Thus, the bulk DOC obtained after filtration and removal of predators and primary producers was mainly refractory, although DOC concentration was high in the surface (Moutin et al., this issue). This surface bulk pool of DOC is probably largely recalcitrant due to UV photodegradation or photooxidation (Keil and Kirchman, 1994; Tranvik and
Stephan, 1998; Carlson and Hansel, 2015) or by action of the microbial carbon pump (Jiao et al., 2010). Such low BGE could be also due to strong resource dependence as low nutrient concentrations cause low primary production rates, and low transfer across food webs. Indeed, Letscher et al. (2015) also observed surface DOC recalcitrant to remineralization in the oligotrophic part of the eastern tropical south Pacific. But as shown by these authors, incubation with microbial communities from the twilight zone, provided by addition of an
inoculum concentrated in a small volume, allowed DOC remineralization. This was explained by relief from micronutrient limitation or potential role for co-metabolism of relatively labile DOC with more recalcitrant DOC. Large stocks of DOC, with C/N ratios ranging 16 to 23 have also been reported in the surface waters of the SPG (Raimbault et al., 2008).

In order to better explain the variability of BGE measurements, we also estimated this parameter
indirectly, using the estimates of community respiration (CR) and BP, when both were measured simultaneously. CR was also estimated along three vertical profiles at each LD site on days 1, 3 and 5, using the mooring lines and 24h- *in situ* incubations as described in Lefevre et al. (this issue). We converted CR to carbon units assuming a respiratory quotient RQ = 0.9, and computed BGE from Ze-integrated BP and CR rates assuming either bacterial respiration (BR) to be within a range of 30 % of CR (BGE=BP/(BP+CR*0.9*30 %)) (Rivkin and
Legendre, 2001; del Giorgio and Duarte, 2002) or 80 % of CR (BGE=BP/(BP+CR*0.9*80 %), Le Mee et al.,





2002; Aranguren-Gassis et al., 2012). The results confirmed the BGE ranges were around those obtained from the biodegradation experiments: 3–12 % at site LDA, 4–17 % at site LDB (with an increasing trend with time from day 1 to day 5: on average 8 % on day 1, 10 % on day 3 and 12 % on day 5) and 2–7 % at site LDC. Overall, the mean BGE was 8 % ± 4 %.

The metabolic state of oligotrophic oceans is still controversial (Duarte et al., 2013; Ducklow and Donney, 2013; Williams et al., 2013; Serret et al., 2015, Letscher et al., 2017), and a consensus emerges that in vitro estimates, (involving $O_2$ derived rates or labelling with $^{18}O_2$ $^{13}C$, or $^{14}C$ isotopes) tend to show net heterotrophy in oligotrophic environments. However, those estimates suffer from many biases related to bottle effects, type of flasks used (selecting light wavelengths), condition of incubations and handling artefacts, as well
as a lack of high frequency measurements. In contrast, *in situ*-based estimates, based on observations of mixed-layer net oxygen exchanges ($O_2$/Ar technique), tend to favour slight net autotrophy (Williams et al., 2013), although these results also suffer from biases, related to assumptions in the mixed layer depth considered and diffusive coefficients used for gases. Another approach based on the use of oxygen sensors in Argo floats recently showed annual NCP close to zero in the South Pacific Ocean (Yang et al., 2017). Recent models
encompassing all seasons and a large areal basis find the global ocean to be net autotrophic, including all five oligotrophic subtropical gyres (Letscher et al., 2017). From the present cruise, Lefevre et al. (this issue) used the in vitro oxygen technique on the mooring lines at the long duration stations and obtained negative NCP at all stations (-96 to -197, -75 to -134 and -60 to -140 mmole $O_2$ m$^{-2}$ d$^{-1}$). Note however the substantial variation which was more or less two-fold at each site. Net heterotrophy was also obtained (although not statistically
different from zero) using in vitro $O_2$ technique analog to ours in the center of the GY, between the Polynesian Archipelago and Easter Island (Van Wambeke et al., 2008b), whereas a non-intrusive bio-optical method showed metabolic balance (Claustre et al., 2008a) in the same area.

Nevertheless, simultaneous estimates of PP, BP and $N_2$Fix rates are almost absent in oligotrophic waters and to date, BP has not been measured in the WTSP. Here we analyze the contribution of primary production to
bacterial carbon demand by comparing them estimated separately in the WTSP using our C-based discrete biological fluxes and considering correction for two biases: the lack of a direct gross primary production (GPP) measurement, and the assimilation of leucine by *Prochlorococcus* in the dark. The ratio of bacterial carbon demand (BCD) to GPP is presented as an index of the coupling between primary producers and heterotrophic bacteria and of metabolic balance (Hoppe et al 2002; Fouilland and Mostajir, 2010): when BCD exceeds GPP,
populations can be temporally non synchronous and/or allochtonous sources of DOM may be required, to sustain heterotrophy.

It is known that the in vitro $^{14}C$ method measure an intermediate state between net PP and GPP. However, Moutin et al. (1999) showed that GPP could be reasonably estimated from daily net PP determined from dusk to dusk as: GPP=1.72 * PP. On the other hand, dealing with the assumptions made to convert hourly
leucine incorporation rates to daily BCD, there are many biases that have been largely debated, including mostly those resulting from daily variability, assumptions on BGE or BR (Alonzo-Saez et al., 2007; Aranguren-Gassis et al., 2012), carbon to leucine conversion factors (Alonso-Saez et al., 2010), and light conditions of incubations including UV (Ruiz-Gonzales et al., 2013). Here, we focus on one largely unexplored bias, that related to the ability of *Prochlorococcus* to assimilate leucine in the dark. Using flow cytometry cell sorting of samples
labelled with $^3$H-leucine during the OUTPACE cruise, Duhamel et al. (in revision) demonstrated the mixotrophic



capacity of *Prochloroccoccus*, as this phytoplankton group was able to incorporate leucine in the light but also, albeit to a lesser extent, in the dark in all examined samples. Nevertheless, this capacity cannot be translated in terms of BP, as to date this group was found to be able to assimilate only few organic molecules, mainly those including N, P or S sources have been examined in different oceanic areas (ATP, leucine, methionine), and more

scarcely glucose as a single C-containing molecule (Duhamel et al., in revision, and ref therein). Using the data acquired at sites LDA, LDB and LDC by means of flow cytometry cell-sorting experiments by Duhamel et al., we calculated an BP corrected ($BP_{corr}$) representing the assimilation of leucine in the dark by heterotrophic bacteria alone. Leucine assimilation by HNA+LNA bacteria corresponded on average to $76 \pm 21$ % (n = 5, range 44–100 %) of the activity determined for the community including *Prochlorococcus* (HNA + LNA + Proc).

Figure 10 presents the distribution of GPP and $BP_{corr}$ along the longitudinal transects under such assumptions. The scale of GPP was adjusted by a factor 100/8 to that of bacterial production, so that with a mean BGE of 8 % the scale of BCD is the same as that of GPP. Assuming this BGE for all our integrated data, GPP is still much lower than $BCD_{corr}$ at some stations (SD4, 5, 6, LDB), but higher at others (SD9, LDA and LDC). Such results confirmed the high variability in the potential contribution of contemporary estimates of GPP to satisfy BCD.

Such comparisons would be more complex if variability of the BGE was included in our results. For example, site LDB illustrates how rapidly these relative fluxes changed during the collapse of the bloom. In relation to the decreasing chl a stocks, decreasing PP, increasing BP, $BCD_{corr}$ to GPP ratios would increase from 1.08 (day 1) to 1.83 (day 5), based on a constant BGE of 8 %. At the same time, though, using the actual BGE increase that we observed from days 1 to 5, the ratio would increase only from 1.08 to 1.2. Unfortunately, it is nearly impossible

to assess all correction and conversion factors at the same scale that we estimated BP and PP, leading to unconstrained budgets (Gasol et al., 2008). LDB was located inside a massive chlorophyll patch, which had been drifting eastwards for several months (de Verneil et al., 2017b) and that could have sustained net autotrophy for long period of times before we arrived to study LDC.

**4.3 Nutrient limitation and relationships with nitrogen fixers**

Nitrogen is primarily limiting bacterial production and primary production in the GY (Van Wambeke et al., 2008a; Halm et al., 2012). It has been shown that labile ammonium and leucine additions could stimulate $N_2$fix rates (Halm et al., 2012). Phylogenetic analyses of the functional gene *nifH* showed prevalence of gamma-

proteobacteria and unicellular cyanobacteria UCYN-A (presumably photo-heterotroph) in the surface layers of the ultra-oligotrophic center of the GY (Halm et al., 2012). However, quantifying gene transcripts in the GY and in the WTSP, Moisander et al. (2014) found *nifH* expression by UCYN-A to be 1–2 orders of magnitude greater than for a gamma proteobacterial diazotroph (γ-24774A11). Along the Australian great barrier reef, as determined by qPCR, the abundance of *nifH* gene copies of Gamma A group (with peaks of only $5.9 \times 10^2$ *nifH*

copies $L^{-1}$) were in general also 1 to 2 orders of magnitude less abundant than those of *Trichodesmium* (Messer et al 2017). Unfortunately, to date, there are no data on the *nifH* gene diversity of heterotrophic nitrogen fxers, nor on their expression, in the euphotic layers for the OUTPACE cruise.

The amount of $N_2$ fixed corresponded to 15–83 % of heterotrophic bacterial nitrogen demand in MA area and 3–35 % in the WGY area. As such, the activity of $N_2$ fixers is not sufficient to sustain all nitrogen needs

of heterotrophic bacteria, whatever are the intermediary processes relating both fluxes, although it can reach a





high participation in the MA area where $N_2$fix rates were greater. Nevertheless, the activity of $N_2$ fixers is responsible for the injection of new N in the microbial food chain, enabling a cascading transfer through non-fixing autotrophs and heterotrophs, and possibly C and N export by sedimentation even in oligotrophic areas of the WTSP (Caffin et al., 2017). The transfer between $N_2$ fixers and heterotrophic prokaryotes occurs on a daily

scale, as confirmed by nano-scale secondary ion mass spectrometry (nanoSIMS) experiments, making it possible to track the fate of $^{15}N_2$ at the individual cell level. A rapid N transfer has been shown from *Trichodesmium* colonies to heterotrophic epiphyte bacterial cells (Eichner et al., 2017). Although it is not known from this study if the $^{15}N$ can reach free living heterotrophic prokaryotes or non-fixing phytoplankton, other experiments suggest that $^{15}N$ fixed by *Trichodesmium* reaches rapidly non-fixing diatoms (Foster et al., 2011; Bonnet et al., 2016).

Using artificial diazotroph cultures inoculated in natural sea waters from the New Caledonia lagoon, Berthelot et al. (2016) showed also a rapid transfer (48h) from a *Trichodesmium* (*T. erythraeum*) and a UCYN-B (*Chrocosphaera. Watsonii*) towards heterotrophic bacteria. Using a qPCR analysis of *nifH* gene copies in selected diazotrophs, focusing on cyanobacteria (unicellular, filaments and heterocystous symbionts) during the OUTPACE cruise, Stenegren et al. (2017) showed that *Trichodesmium* dominated in surface layers (0–35m)

within MA, but was rare or undetected in the WGY. Within MA the second most abundant populations of cyanobacteria detected were UCYN-B, and the third heterocystous endosymbionts (diatom-diazotroph associations). Among the investigated cyanobacterial diazotrophs, this study showed a temperature - depth gradient separating two groups of cyanobacteria, *Trichodesmium* occupying the warmest and shallowest waters, and UCYN-A occupying the coldest and deeper waters while UCYN-B was more widespread. Recent evidence

suggests also a rapid transfer from the symbiotic, photo-heterotrophic cyanobacterium UCYN-A, which have much greater growth rates than *Trichodesmium* to its associated eukaryotic algae (Martinez-Perez et al., 2016).

   Consequently, most $N_2$ fixing cyanobacteria detected in the WTSP during this cruise have a potential to transfer N rapidly (1–2 days) towards strict heterotrophic bacteria or non-fixing phytoplankton. We found that the relationship between $N_2$fix rate and BP was higher within the mixed layers and when the $T_{DIP}$ is low. Low

$T_{DIP}$ indicates phosphate deficiency (Moutin et al., 2008). Areas with low $T_{DIP}$ are thus reflecting intense competition for the available DIP source, a competition between $N_2$ fixers, where *Trichodesmium* dominates, heterotrophic prokaryotes and primary producers. From our results in the layers with low $T_{DIP}$, which are characterized by a better correlation between BP and $N_2$fix than between BP and PP, it seems that both $N_2$ fixers and heterotrophic prokaryotes are better competitors for DIP than other primary producers. Indeed, at the site

LDB within the mixed layers, BP is increasing after N addition alone but also after P addition alone, which suggest a direct N limitation of BP in the first case and a cascade effect after P addition in the second case, stimulating $N_2$ fixers and a new, rapid transfer of N and labile C through the food web, then available to stimulate BP. At sites LDA and LDB, the addition of the 3 elements NPG stimulated BP more than P alone or N alone, suggesting possible NP co-limitation of heterotrophic prokaryotes. Furthermore, if N was shown to be the

first limiting nutrient during short time scale experiment, addition of P stimulates $N_2$fix, PP and export at larger time scales (Van Den Brock et al., 2004; Berthelot et al., 2015, Gimenez et al., 2016).

   Below the surface layers, where $T_{DIP}$ increases, and where UCYN-A dominates, DIP becomes available, and nitrate diffusing through the nitracline sustains primary production, BP and PP are correlated, suggesting a strong coupling between BP and PP through the release of labile organic C source: within these layers BP is

limited fist by labile C, secondarily by N (Fig. 9).





Finally within the WGY area, where $T_{DIP}$ are higher in the mixed layer (more than 100 h), with detectable DIP concentrations reaching 100 nM, the activity of $N_2$ fixers was extremely low. In this area, BP was limited mainly by the availability of energy or labile C. Indeed, within site LDC glucose alone stimulated BP on a larger extent than did N alone, even when the latter was provided in the form of ammonium. UCYN-B

dominated the $N_2$ fixing populations in this area, which accounted for 81−100 % of the total detected cyanobacterial *nifH* gene copies (Stenegren et al., 2017). Among UCYN-B, *Crocosphaer*a is one of the most studied representatives. One of its sub-populations is recognized to produce EPS (Bench et al., 2016), which could be a significant energy source for heterotrophic prokaryotes. In the North Pacific Subtropical gyre, Wilson et al (2017) hypothesized that *Crocosphaer*a could fix $N_2$ in excess of its growth requirements and could leak

fixed N from the cells. They also showed a highly dynamic of *Crocosphaer*a growth and decay during diel cycles survey, suggesting efficient mortality sources. The resources provided by leakage, lysis and grazing process likely direct energy and N towards heterotrophic bacteria at a daily scale when $N_2$ fixation is favourable, as in the North Pacific. However in the WGY, rates of $N_2$fix rates are very low and sustain a low percentage of bacterial and phytoplankton N demand.


### 5    Conclusion

Our results provided a unique set of simultaneous measurements of BP, PP and $N_2$fix rates in the WTSP, which together, explained competition between primary producers and heterotrophic prokaryotes for

nitrogen. In surface, nitrogen and relatively DIP depleted waters, BP was more strongly related with $N_2$fix than with PP, while the more traditional coupling of BP with PP occurred deeper in the euphotic zone. This suggested an efficient superiority by small-sized heterotrophic prokaryotes in comparison to the non diazotrophic phytoplankton for the N and P availability. However, a rapid transfer can occur through the microbial food web, as suggested by other studies using nanoSIMS technique in the same area. The rapidity at which the bloom

collapsed at LDB during the 6 days site survey demonstrates the necessity for studying the fate of PP in the oligotrophic ocean across larger temporal and spatial scales. We show that the interpretation of fluxes based on instantaneous methods (radioisotopic labelling) needs regular tests to verify the major methodological hypothesis, in particular, the use of the leucine technique to estimate BP should be used with caution in N-limited environments due to the potential mixotrophy by cyanobacteria. The variability of bacterial carbon

demand and gross primary production rates measured in this study reveal the highly diverse metabolic status of the WTSP.

*Acknowledgements.*

We thank S Helias, O Grosso, and M Caffin for their support in providing nutrient and $N_2$ fixation data, N Bock for linguistic check, S Bonnet and M Benavides for scientific advice on diazotrophs. This is a contribution of the OUTPACE (Oligotrophy from Ultra-oligoTrophy PACific Experiment) project (https://outpace.mio.univ-amu.fr/) funded by the French research national agency (ANR-14-CE01-0007-01), the LEFECyBER program (CNRS-INSU), the GOPS program (IRD)  the CNES (BC T23, ZBC 4500048836) and

the European FEDER Fund (project 1166-39417). The OUTPACE cruise (http://dx.doi.org/10.17600/15000900)




was managed by MIO (OSU Institut Pythéas, AMU) from Marseilles (France). SD was funded by the National

Science Foundation (OCE-1434916).

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



Table 1. Review of Integrated primary production rates published in the South Pacific, PP fluxes in mg Cm$^{-2}$ d$^{-1}$. Only open sea data were included.

| Reference | Lat | Long | Area | Number of stations | Period | Technique | PP fluxes |
|---|---|---|---|---|---|---|---|
| Data from Moutin in Van Wambeke et al, 2008a | 8–13°S | 140–130°W | Marquesas archipelago | 5 | Nov-Dec 2004 | 14C, 0.2 µm polycarbonate, deck incubations | 250–680 |
| | 15–30°S | 130–100°W | center GY | 11 | | | 76–167 |
| | 30–33°S | 95–78°W | East GY | 6 | | | 195–359 |
| Rii et al, 2016 | 25–26°S | 104–100°W | Center GY | 2 | Nov-Dec 2010 | 14C, 0.2 µm polycarbonate, deck incubations | 216–276 |
| | 23°S | 88°W | East GY | 1 | | | 600 |
| Halm et al, 2012 | 23–27°S | 165–117°W | center GY | 7 | Dec 2006-Jan2007 | 13C, GF/F, deck incubations | 8–33 |
| | 38–41°S | 153–133°W | Southern rim of the GY | 3 | | | 79–132 |
| Menkes et al, 2015 | 17–23°S | 157–170°E | New Caledonian Exclusive Econ. Zone | 7 | July-Aug 2011 | 14C, 0.4 µm polycarbonate, Pvs I curves | 352 ± 160 |
| | | | | 5 | nov-dec 2011 | | 231 ± 133 |
| Young et al, 2011 | 27–29°S | 160–162°E | Eastern Australia off shore | 2 | | 14C, GF/F filters, deck incubations | 260–910 |
| Ganachaud et al, 2017 | 3–9°S | 146–152°E | Solomon Sea | | Feb/March 2014 | 13C, GF/F | 204–1116 |
| | 5–12°S | 147–165°E | | | June/Aug 2012 | , deck incubations | 480–1200 |
| Moutin et al 2017, this study | 18–19°S | 159°E–170°W | Melanesian Archipelago | 14 | Feb/March 2015 | 14C, 0.2 µm polycarbonate | 148–858 |
| | 18°S | 169–149°W | Western GY | 4 | | , deck incubations | 55–208 |





Table 2. Physical and biological characteristics of some stations sampled during the OUTPACE cruise. Depth of the dcm (deep chlorophyll maximum, based on vertical profiles of in vivo fluorescence), $\sigma_t$: sigma-theta at the dcm (kg m$^{-3}$), Ichl a (integrated chlorophyll a from fluorimetric discrete analyses), IN$_2$ fix (integrated N$_2$ fixation rates), IPP (integrated primary production), IBP (integrated bacterial production, at the depth of the euphotic zone). WMA area comprised station SD1, 2, 3 and LDA, EMA area comprised SD 6, 7, 9 and 10. WGY area comprised SD13, 14, 15 and LDC. In order to encompass only spatial variability for WMA, EMA and WGY areas, means and ranges of dcm depths and of $\sigma_t$ at the dcm were based on the averages values set individually at each SD or LD stations as more than one cast was made at each station. Means and range values given for LDA, LDB and LDC illustrate the temporal variability at LD sites: all ctd casts sampled at each LD site down to 200 m were included.
* values from Bonnet et al. (this issue) and Caffin et al. (2017) also presented in Fig. 3a
** values from Moutin et al. (this issue) also presented in Fig. 2 and Fig. 3a
*** values from Moutin et al. (this issue) also presented in Fig. 2 and Fig. 3b
at station SD13, BP and N$_2$fix rates were not measured; PP obtained was abnormally low (55 mg C m$^{-2}$ d$^{-1}$) and excluded from the mean

| | | WMA | EMA | WGY | LDA | LDB | LDC |
|---|---|---|---|---|---|---|---|
| dcm depth | mean ± sd (n) | 82 ± 10 (4) | 105 ± 10 (4) | 136 ± 14 (4) | 81 ± 9 (46) | 50 ± 18 (47) | 131 ± 7 (46) |
| m | range | 72 – 91 | 91 – 115 | 123 – 154 | 63 – 101 | 10 – 77 | 115 – 154 |
| $\sigma_t$ at the dcm | mean ± sd (n) | 23.8 ± 0.4 (4) | 24.2 ± 0.3 (4) | 24.53 ± 0.09 (4) | 23.55 ± 0.05 (46) | 23.1 ± 0.7 (47) | 24.62 ± 0.02 (46) |
| kg m$^{-3}$ | range | 23.5 – 24.3 | 23.8 – 24.6 | 24.4 – 24.6 | 23.47 – 23.64 | 21.7 – 23.9 | 24.59 – 24.67 |
| Ichl a | mean ± sd (n) | na | 28.7 ± 6.2 (4) | 18.1 ± 4.5 (4) | 26.0 ± 2.6 (5) | 38.9 ± 10.4 (5) | 16.2 ± 1.3 (7) |
| mg Chl a m$^{-2}$ | range | na | 23.6 – 37.8 | 13.2 – 23.6 | 23.7 – 29.6 | 23.9 – 53.2 | 14.0 – 17.7 |
| IN$_2$ fix (deck) | mean ± sd (n) | 0.65 ± 0.21 (4)* | 0.50 ± 0.27 (4)* | 0.09 ± 0.08 (3)*** | 0.63* | 0.94* | 0.07* |
| nmole N m$^{-2}$ d$^{-1}$ | range | 0.48 – 0.96 | 0.21 – 0.85 | 0.02 – 0.17 | | | |
| IN$_2$ fix (in situ) | mean ± sd (n) | | | | 0.59 ± 0.05 (3)* | 0.70 ± 0.30 (3)* | 0.06 0.01 (3)* |
| nmole N m$^{-2}$ d$^{-1}$ | range | | | | 0.53 – 0.63 | 0.38 – 0.98 | 0.05 – 0.08 |
| IPP$_{deck}$ | mean ± sd (n) | 481 ± 147 (4)*** | 471 ± 276 (4)*** | 154 ± 55 (3)*** | 698* | 383* | 213* |
| mg C m$^{-2}$ d$^{-1}$ | range | 367 – 698 | 192 – 853 | 104 – 213 | | | |
| IPP$_{in situ}$ | mean ± sd (n) | | | | 267 ± 79 (3)* | 436 ± 72 (3)* | 155 ± 8 (3)* |
| mg C m$^{-2}$ d$^{-1}$ | range | | | | 200 – 354 | 361 – 507 | 149 – 165 |
| IBP within Ze | mean ± sd (n) | 99 ± 15 (4) | 95 ± 12 (4) | 33 ± 2 (3)*** | 98 ± 16 (5) | 113 ± 15 (6) | 45 ± 5 (6) |
| mg C m$^{-2}$ d$^{-1}$ | range | 82 – 120 | 80 – 110 | 31 – 35 | 81 – 115 | 86 – 133 | 38 – 50 |





**Table 3**. Results of multiple regressions log BP=f (log PP, log $N_2$fix). BP Units before log- transformation is ngC $l^{-1}$ $h^{-1}$. Y int : Y intercept.

| Units before log transformation | | mgC $m^{-3}$ $d^{-1}$ | nmole N $l^{-1}$ $d^{-1}$ | | | |
|---|---|---|---|---|---|---|
| independent variables | | PP | $N_2$fix | Y int | n | r |
| $T_{DIP} \leq 100h$ | part coeff ± sd | 0.23 ± 0.11 | 0.38 ± 0.10 | -0.56 | 36 | 0.589 |
| | t (p) | 2.04 (0.02) | 3.82 (0.0002) | | | |
| $T_{DIP} > 100h$ | part coeff ± sd | 0.43 ± 0.08 | 0.09 ± 0.05 | -0.47 | 51 | 0.66 |
| | t (p) | 4.91 (< 0.0001) | 1.82 (ns) | | | |

**Table 4**. Results of multiple regressions log $T_{DIP}$=f (log BP, log PP, log $N_2$fix). Units of $T_{DIP}$ before log-transformation is h. Y int : Y intercept.

| Units before log transformation | | ngC $l^{-1}$ $h^{-1}$ | nmol N $l^{-1}$ $d^{-1}$ | mgC $m^{-3}$ $d^{-1}$ | | | |
|---|---|---|---|---|---|---|---|
| independent variables | | BP | $N_2$fix | PP | Y int | n | r |
| All data | part coeff ± sd | -1.06 ± 0.2 | -0.23± 0.07 | -0.18 ± 0.15 | 3.82 | 91 | 0.81 |
| | t | -5.2 | -3.1 | -1.2 | | | |
| | (p) | (p< 0.0001) | (p=0.0027) | (ns) | | | |
| Depth ≤ 20m | part coeff ± sd | -1.11 ±0.34 | -0.07 0.13 | 0.54± 0.27 | 3.88 | 47 | 0.76 |
| | t | -3.2 | -0.5 | 1.97 | | | |
| | (p) | (p= 0.0024) | (ns) | (ns) | | | |

**Table 5**. Results of biodegradation experiments. Growth rates determined from BP data, degradation rates computed from DOC data and BGE computed from eq 1.

| | LDA | LDB | LDC |
|---|---|---|---|
| growth rates ($h^{-1}$) | 0.33 ± 0.05 | 0.08 ± 0.02 | 0.14 ± 0.02 |
| degradation rates ($d^{-1}$) | 0.039 ± 0.002 | 0.07 ± 0.007 | 0.012 ± 0.003 |
| initial DOC stock (μM) | 83 | 83 | 75 |
| % labile DOC | 5.3 | 5 | 2.4 |
| BGE (%) | 12.9 | 6.3 | 6.7 |





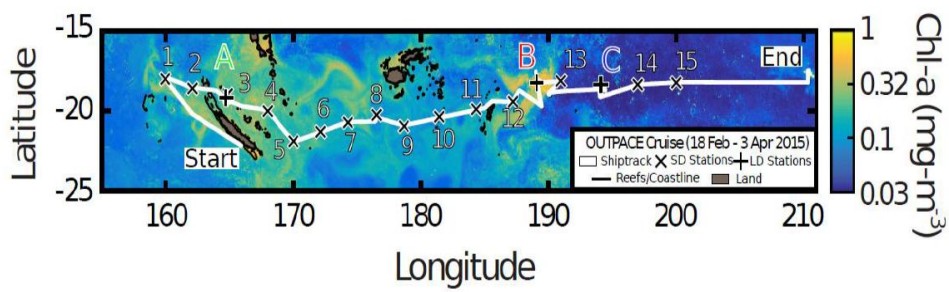

**Figure 1** Quasi-Lagrangian Surface Chlorophyll-a concentration (mg m⁻³) during the OUTPACE cruise. The satellite data are weighted in time by each pixel's distance from the ship's position for the entire cruise. The white line shows the vessel route (data from the hull-mounted ADCP positioning system). Coral reefs and coastlines are shown in black, land is grey, and areas of no data are left white. The positions of the short (long) duration stations are shown by cross (plus) symbols. The ocean color satellite products are produced by CLS.
Figure courtesy of A. de Verneil (02/06/2017).





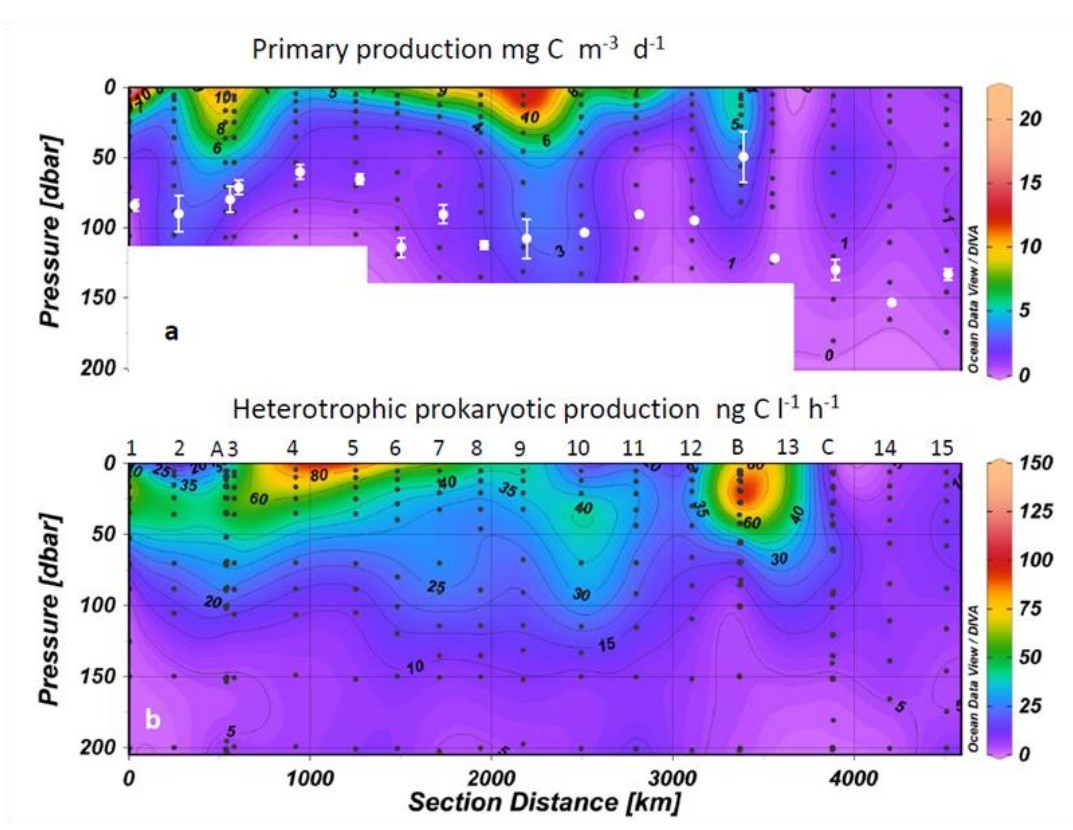

**Figure 2** Distribution of primary production (a) and heterotrophic prokaryotic production (b) along the OUTPACE cruise transect. Interpolation between sampling points in contour plots was made with the Ocean Data View software (VG gridding algorithm, Schlitzer, 2004). In order to be homogeneous for the whole transect, for sites LDA, LDB and LDC the data plotted for PP was from a single profile, that of $PP_{deck}$, while for BP we plotted all profiles. The white dots in (a) correspond to the average ± sd of the dcm depth at each station. The white rectangles mask abnormal extrapolation due to the absence of PP data





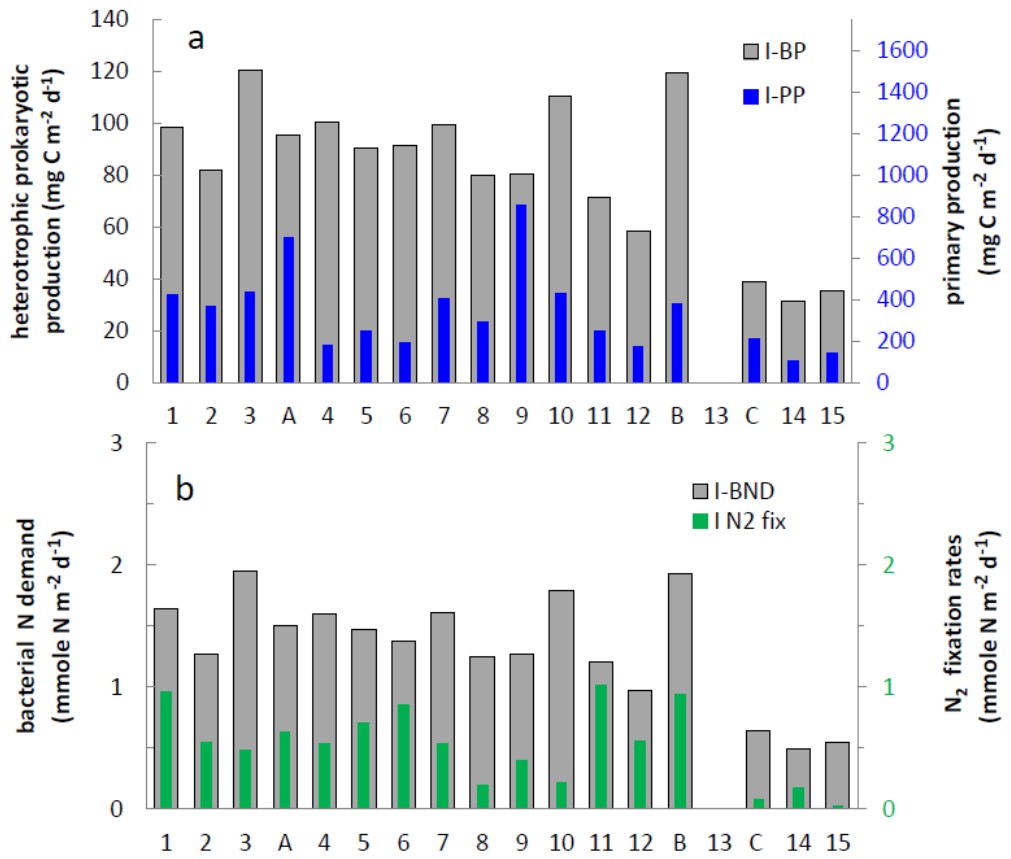

**Figure 3** a) Histogram distribution of integrated heterotrophic prokaryotic production (IBP) and primary production (IPP$_{deck}$) along the transect, data were integrated down to the euphotic zone. b) Histogram distribution of bacterial nitrogen demand (assuming a bacterial C/N ratio of 5 and no nitrogen excretion) and N$_2$ fixation rates along the transect, data were integrated down to the deepest sampled depth for N$_2$ fixation rates. Data plotted for sites LDA, LDB and LDC correspond to BP, PP$_{deck}$ and N$_2$fix measured on day 5.

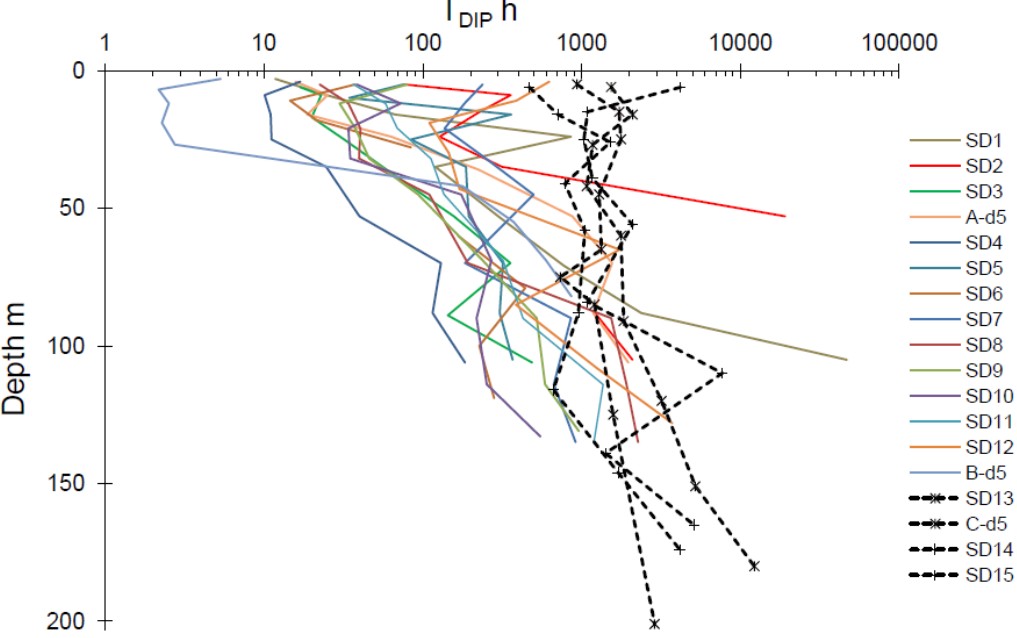

**Figure 4** Vertical distributions of phosphate turnovertimes ($T_{DIP}$). Dotted profiles correspond to SD13, 14 15 and site C.





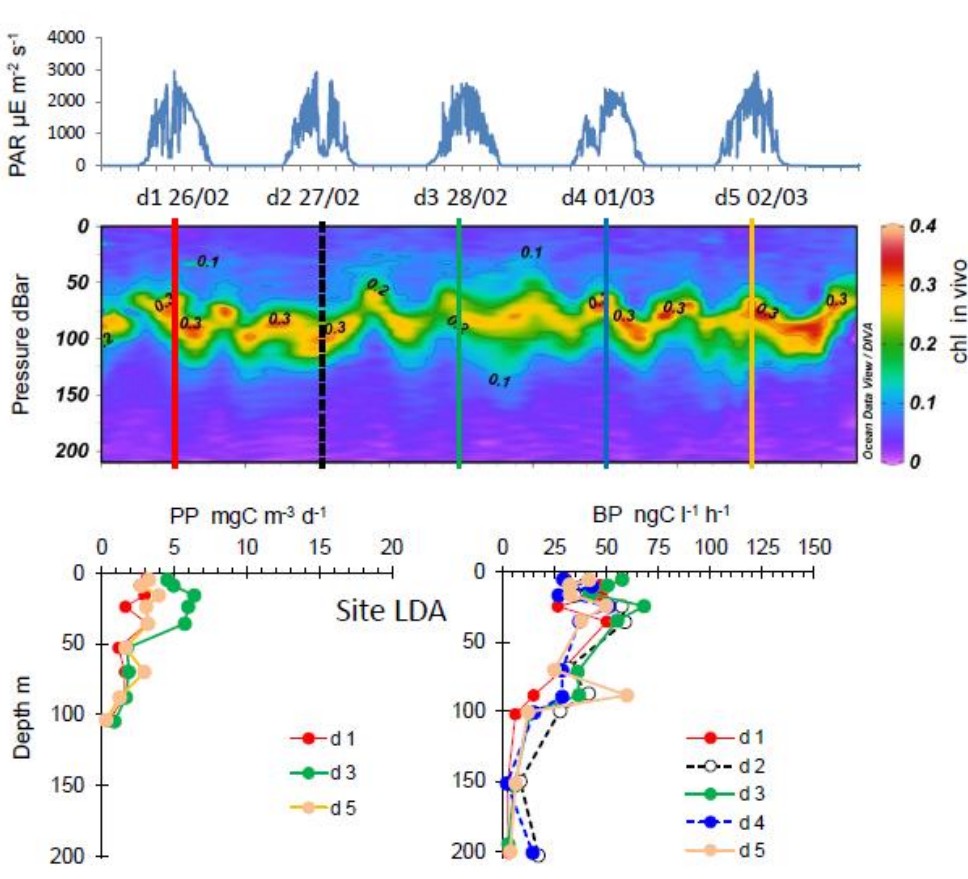

**Figure 5** Evolution of surface PAR, chl in vivo, PP and BP at the site LDA. Time units in local time, day1 was February 26, 2015. BP samples were taken at the 12:00 ctd cast, while samples for PP$_{in\ situ}$ were taken at the 3:00 ctd casts (day 1, 3 and 5).





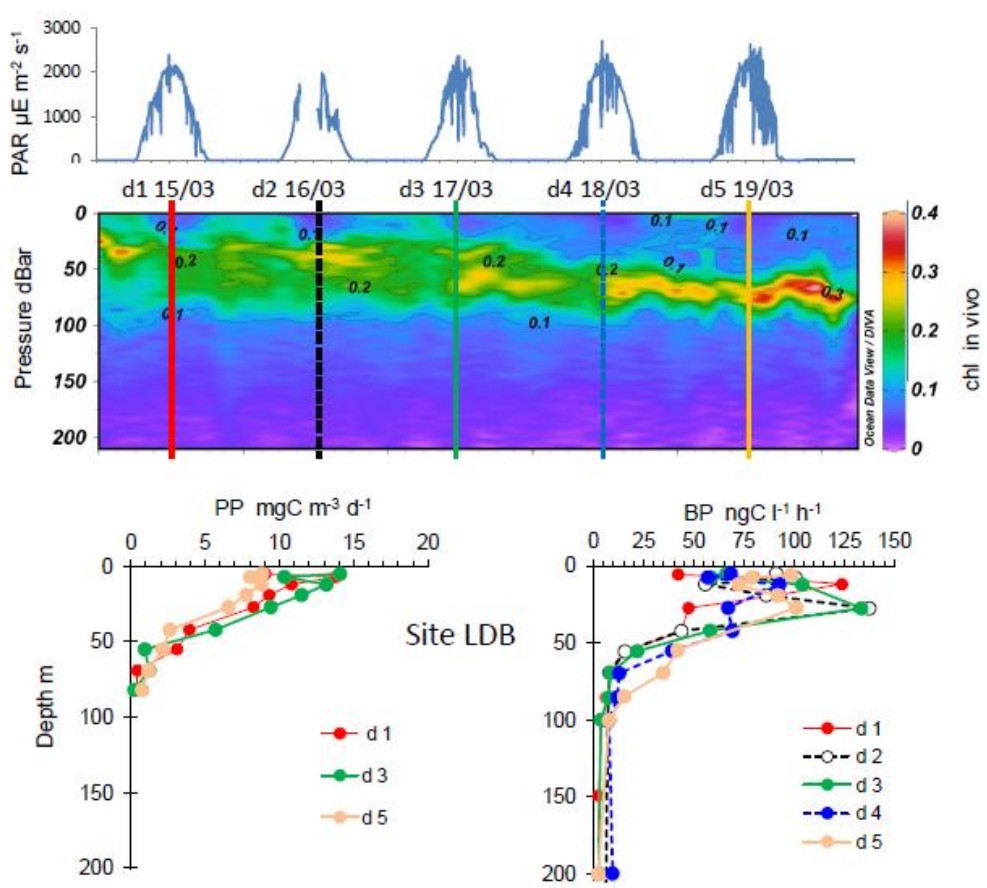

**Figure 6** Evolution of surface PAR, chl in vivo, PP and BP at the site LDB. Time units in local time, day1 was March 15, 2015. BP samples were taken at the 12:00 ctd cast, while samples for $PP_{in\ situ}$ were taken at the 3:00 ctd casts (day 1, 3 and 5).



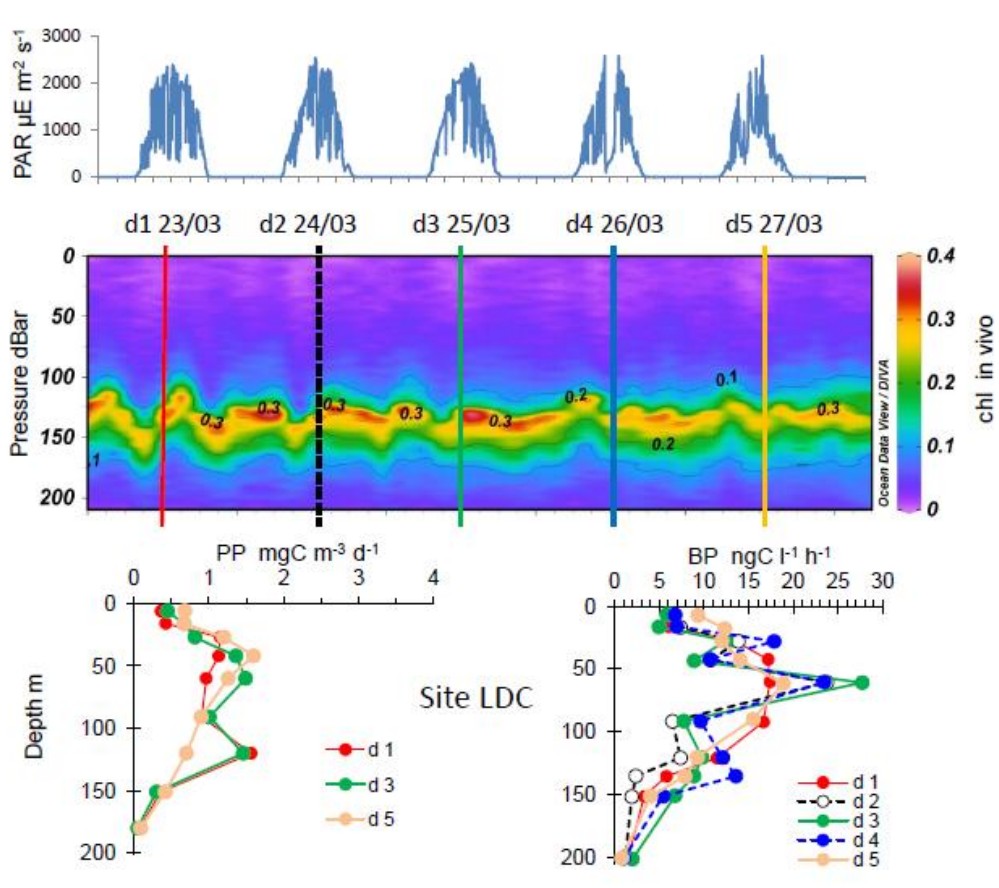

**Figure 7** Evolution of surface PAR, chl in vivo, PP and BP at the site LDC. Time units in local time, day1 was March 23, 2015. BP samples were taken at the 12:00 ctd cast, while samples for PP$_{in\ situ}$ were taken at the 3:00 ctd casts (day 1, 3 and 5).





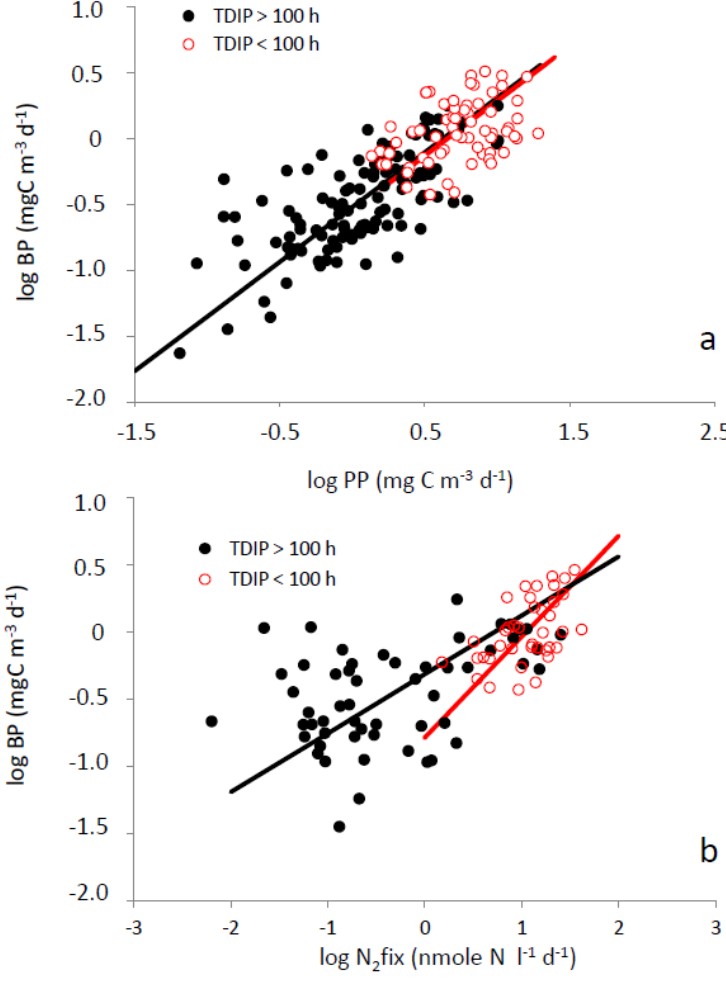

**Figure 8** Log-log relationships between volumetric rates of heterotrophic prokarytotic production (BP) and primary production (PP, panel a) and between BP and nitrogen fixation rates ($N_2$fix, panel b). Red and black dots show samples where $T_{DIP}$ were above and below 100 h, respectively. Lines are fitted Tessier model II regressions for data clustering samples where Tdip values were higher (black lines) and lower (red lines) than 100 h.





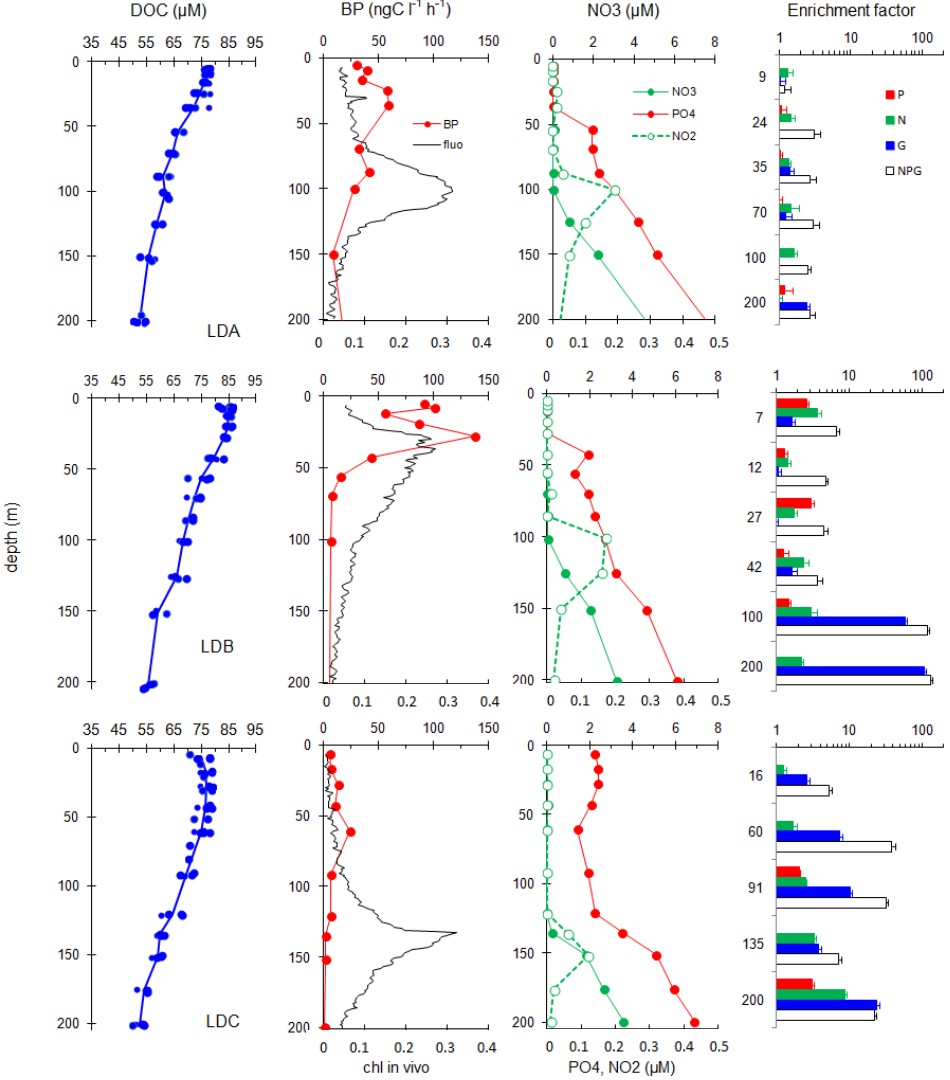

**Figure 9** Enrichment experiments. Initial conditions illustrated by vertical profiles (0-200 m) of in vivo fluorescence, BP, nutrients (nitrate (NO3), nitrite (NO2), ammonium (NH4) and DIP (PO4)) and enrichment factors sampled from the 12:00 CTD cast on day 2 of occupation at each LD site. As DOC was not sampled on this cast, we showed the data from all the other casts at the corresponding LD site (dots) and the average profile (line). Enrichment factors are the ratio of BP after a given enrichment (DIP: P in red; nirate+ammonium :N in green, glucose: G in blue, and all componenrs: NPG in black) compared to the unamended control, both measured 24 h after incubations. The error bar is sd within triplicates, and a bar is shown only if BP is significantly higher than in the control (Mann-Whitney test, $p < 0.05$).





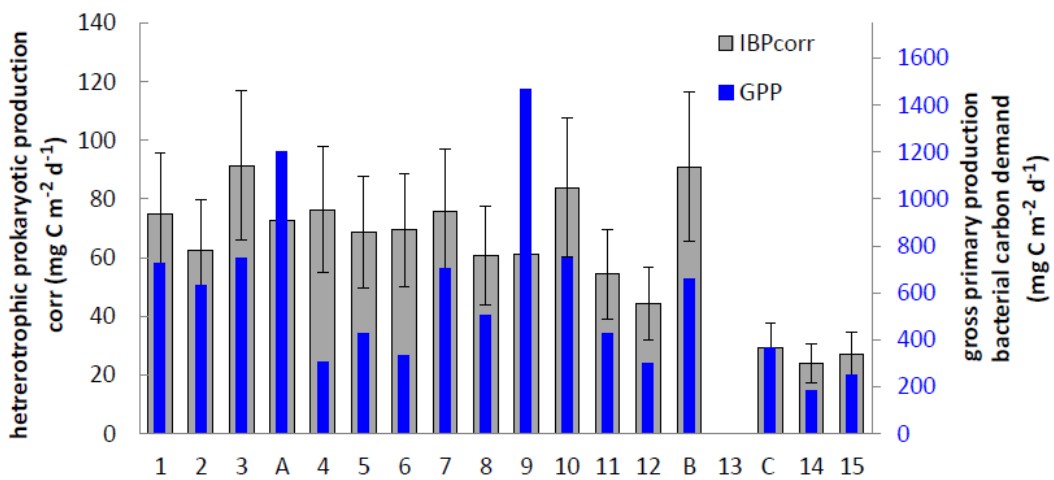

**Figure 10** Histogram distribution of integrated heterotrophic prokaryotic production corrected for *Prochlorococcus* assimilation (IBP$_{corr}$, grey bars, left scale) and gross primary production (GPP, blue bars, right scale) along the transect. Scales of BP$_{corr}$ and GPP are proportional by a factor 100/8 so that GPP scale is also that of BCD, and thus blue bar higher than grey bar means GGP higher than bacterial carbon demand.