# Peer review of "Dynamics and controls of heterotrophic prokaryotic production in the western tropical South Pacific Ocean: links with diazotrophic and photosynthetic activity."

_Biogeosciences, 2017_

## Referee Comment (RC1) · M. Aranguren-Gassis (Referee) · 31 Jan 2018

General comments

In the manuscript "Dynamics of phytoplankton and heterotrophic bacterioplankton in the western tropical South Pacific Ocean along a gradient of diversity and activity of diazotrophs", Van Wambeke et al. present estimations of heterotrophic prokaryotic production in the Western Tropical South Pacific region, and they explore the causes of its variability, focusing on autotrophic activity and nutrient availability. The data pre-

sented in the paper are very valuable, as the carbon budget in the oligotrophic regions is still a topic of debate, in part because of lack of data to adequately characterize the bacterial contribution to it. The data presented are a great mix of observations and experiments, and the analysis made have great potential. However, the way the paper is written makes the information confusing and the conclusions vague. The title doesn't reflect the contents of the paper, the methods are not complete, the discussion is not well structured or clear, and the conclusions are not in line with the results highlighted in the discussion (See details in specific comments). The English and the writing needs profound review.

Specific comments

- Title: The authors mention in the title a gradient of diversity and activity of diazotrophs, but such gradient is not shown in the paper. They base part of their discussion in different groups of diazotrophs described during the same cruise by other authors (lines 530-541), but as they described it, it is not a diversity gradient, just differences in the dominant genera. The diazotrophs activity gradient is not clear either in the paper. I would suggest for the title to focus more on the analysis made to elucidate the factors controlling the bacterioplankton activity in different regions of an oligotrophic system.

- Abstract: The abstract is a good summary of the paper, but I think some parts can be removed: * Line 28: the i.e. can be removed, it makes the sentence too long, and it is not necessary * Line 30: The BGE estimation doesn't provide useful information here * Lines 33-36: I don't find this information about the bloom developed along the paper, I think this should be removed from here.

- Methods: Some variables have a lot of weight in the discussion but methods are not described. For example Nitrogen fixation rates, community respiration and GPP, or nutrients (nitrate, nitrite, and phosphate) concentrations. At least a brief description of the methods should be included, even if they have been explained somewhere else.

- Lines 109-112: The criteria used for stations selections is not stated. Even if it is

described in other papers, a better explanation should be included here because it can affect the interpretation of the results.

- Line 113: It should be mention here that the sampling in LD stations was Lagrangian.

- Lines 151 and 161: The word "occasionally" is too vague, please specify at least how many times.

- Lines 165-179: The incubation time used should be specified.

- Lines 185 and 207: Are the cast numbers necessary? I think they can be removed.

- Line 230: The authors talk about a gradient, but they don't specify what kind of gradient. A gradient of productivity? A gradient of diazotrophs activity?

- Lines 240-245: Authors say that nutrients and organic matter distribution allowed them to distinguish two regions, but those data are not shown in the paper. At least, a description of the differences between the regions should be included. For example, you can include some extra data in Table 2 with nutrients and organic matter concentration, or whichever criteria used to identify those two regions.

- Lines 236-246 are confusing and need to be rewritten.

- Line 249: "Averaged per SD station, the dcm fluctuated..." I don't understand what that means and what the following ranges refer to.

- Line 269: I suppose what authors meant is that TDIP increased with depth, not the vertical profiles themselves.

- Line 281: I don't think the periodicity of the dcm fluctuation is evident in figure 5. That fluctuation and the increase of fluorescence in the afternoon are not well described. Those patterns are not apparent in the figures, and not statistical analysis is presented. However, I think those results can be avoided because authors don't mention them along the discussion at any point.

-Line 286: Values from In situ and on deck incubation cannot be directly compared, as temperatures for the incubations are different. Particularly for samples from depths below the mixer layer. So authors shouldn't highlight a higher value from on deck incubations without a proper analysis of correspondence between on deck and in situ estimations.

-Lines 324-326: Figure 8 shows a linear regression analysis, but in the text, authors present the correlation coefficient (r), but not in the figure or in the text they mention the significance of the fitting. With the correlation coefficient, authors shouldn't interpret the results as a dependency, because the correlation between two data sets doesn't indicate causality of one of the variables from the other.

- Line 331-334: Those N2fix temporal trends are not shown anywhere. If authors don't want to show them in a figure, use some statistics to state those patterns.

- Line 393-404: The definition of the regions is not well described. Some suggestions: * Use the same terms to name them along the entire manuscript. * Show them on the map (figure 1) * Refer here to Table 2, and complete the table with the criteria described here (nitracline depth). * Define what "distinct nutrient distributions" means or show it in a plot * In figure 4 put in different colors or patterns the profiles for each region

- Line 422: "after filtration and removal... producers" corresponds to methods section.

- Line 431-432: Why is the information in the last sentence of this paragraph relevant? Please, elaborate

- Lines 445-462: This paragraph is not well linked to the rest of the discussion. There is not any mention of the present paper results, and it is not clear the contribution of the present paper to the debate described in the paragraph. Please, elaborate.

- Lines 482-488: This paragraph is confusing and needs rewriting

- Lines 490-492: I don't understand the calculations described in here

[Figure]

- Lines 490-500: What is the overall contribution of all these calculations to the paper?

- Lines 506-509: I don't see the relation between the three first lines of the paragraph and the following ones.

- Lines 548-549: I think it is incorrect to deduce competition ability from this correlation

- Conclusions: The conclusions don't reflect the discussion or even the results presented

- References: There are six not published references in the list, and some of them have data with a lot of weight in the discussion.

- Table 1: Better put the PP units on the table. It would be really helpful to group the rows by region, so it is easier to follow the description in Lines 403-415.

- Figure 1: a more general map to locate the cruise area would be useful. In the legend, the fourth line will be easier to read using "respectively".

- Figure 2: I suggest using the same units for the two panels to make them directly comparable. Put stations number in both plots.

- Figure 3: * The units on the panel B are incorrect, it is mmol. * Why are data at station 13 missing? I don't find the explanation. * As I understand, the bars on figure 3 represents the values of each variable in every station, so these plots are not histograms, they are bar charts. * Bars should be represented by the corresponding error bars.

-Figure 4: this figure needs to be improved. Some suggestions: * Make the station names consistent with other figures. * Group the profiles by region, with color or pattern, or make 3 panels, one for each region. * Talking about this figure in the text (line 269) authors use the phosphacline. Represent the phosphacline here to help. * I would incorporate the profiles for the Long stations in figure 9, as you use the information in the discussion of the experiments.

- Figures 5, 6 and 7: * Vertical axis in the second panel should say fluorescence instead

of chl, as you explain in the text (e.g., line 280) * When describing these figures in the text, you use the density. Including the density level lines in the figures will be a good idea. * Explain in the legend what the vertical lines in the second panel mean, and make them the same color than the profiles in the third panel.

- Figure 10: What is it that you represent in here is not clear for me, not in the text or the figure legend. Please make the description of the calculation more clear and explain better the meaning and the interpretation. Use station names consistent with the rest of the paper.

Technical corrections - Line 23: space is missing between With and N2 loom - Lines 94 and 97, the period is missing - Lines 110 and, the first n in Lagrangian is missing - Line 119: I think authors meant experiments (in plural) - Line 234: a bracket is missing before the references - Lines 236-246 are confusing and need to be rewritten. - Line 249: "Averaged per SD station, the dcm fluctuated..." I don't understand what that means and what the following ranges refer to. - Line 269: Vertical profiles cannot increase or decrease with depth, I assume you are talking here about TDIP decreasing with depth. Please rewrite. - Line 278: Please, check this sentence. Space is missing between down and in. "Comumn" I suppose means columns with l. There is a comma instead of a period after the bracket. The sentence doesn't make sense in general. - Line 287: delete the bracket before IPPDECK - Line 327: PP instead of BP - Line 345: Those abundances, I guess are bacterial abundances - Line 346: a decimal point is missing in 014 - Line 347: the lowest instead of the lowerest instead of the lower

---

## Referee Comment (RC2) · Anonymous Referee #2 · 12 Feb 2018

In this article the authors present the results from a study into bacterial and primary production in the tropical south Pacific ocean. The paper fits perfectly within the scope of Biogeosciences. I found the article interesting to read with some very interesting insights into the carbon balance of this part of the Tropical South Pacific, a region that has been rather less studied than some of the other oceanic provinces.

While the actual methods used can be considered as relatively classic in the domain, their application to this little studied area is novel. Indeed, although several authors have worked in the Tropical South Pacific, the vast majority of these studies have looked

at either N2 fixation alone or have been conducted in the coastal areas near to Islands. This data from the open ocean is particularly interesting and novel. The assumptions of the methods are appropriate and are clearly outlines.

I am wondering why was the ratio 400ml of bacterial 'inoculum' chosen for addition to 2.6L?

The conclusions are approriate and provide some interesting insigths into what is limiting bacterial production in this part of the ocean. Notably, it appears that available N is the limiting factor - which of course underlines the importance of N2 fixing organisms in this environment, as has been already shown by other work from this group.

I was a little perplexed as to why some results were shown in the methods section Pg 4, line 135.

The results section is sufficient to support the conclusions - I have one comment here though - it was a little awkward to have quite a few associated datasets were in other articles - it was a bit difficult to do a "stand-alone" review. But the authors do clearly give credit for other work and they clearly indicate what are their new additions.

The experiments and calculations are well described and will allow for replication by other scientists.

The Title clearly reflects the contents, particularly if we take into account the whole group of papers from the Outpace experiment.

The abstract is clear but I wondering if the last sentence should not appear earlier in the text, it does seem to be a little be deconnected from the rest of the text. Perhaps the authors can rephrase it if they wish to leave it as a last sentence or move it up.

Yes, the article is well structured, clear and I really enjoyed reading it. The language is fluent and clear and the appropriate formulae and correction factors used are presented clearly when needed.

concerning the tables : Table 1 and 2 : both of these tables are a little blurry - maybe check that in a revised version? Also, can the authors add the units into the table (I know they are in the legend, but I always find it easier to follow when they are in the table itself). Table 5 : can the authors add in if its the mean +/- the SD or the SE? Figure 1 : is a little hard to see - but maybe it's my printout - nevertheless, can the authors check that the figure is clear and not blurry. Figure 3 : check the format of the legend titles (add in uppercase letters when needed). Figure 4: why did the authors choses to put in the black dotted lines? It rather draws the eye at the cost of the other profiles.

Overall, can the authors unify the format of the axis titles on the figures - some have () some do not. Also can they check the clarity of the contour maps and the colour of the words/numbers on the graphics - sometimes they are hard to read (see figs. 5-7b).

pg 9 line 341 : non signficant for PP

line 250 : what do the authors mean here 'determined by fluorometry'? Don't both methods employ fluorometrey (Turner vs CTD)?

pg 11, line 420 : 6-12% is not that low. line 440 : check spelling of Lemee here (it's ok in the Refs).

Paragraph starting 455: negative NCP values have also been observed in the oligotrophic water off-shore of New Caledonia (Pringault et al. Biogeosciences 2007. I agree with the authors that calculating up hourly incubation values to daily ones is fraught with errors. Do the authors have an estimate of how much error may be introduced from these factors? It is interesting to note that Prochlorococcus could be responsible for up to 56% of leucine uptake - this could have some very strong implications for BCD calculations and hence, ecosystem metabolism calculations in areas where Prochlorococcus is abundant. What about the diazotrophs? Do they take up leucine? Is there any information on this?

530 : what is an artificial diazotroph culture? 570: not sure what the authors mean here in the sentence starting "They also showed..." - can the authors revised this? 589: what do the authors mean by "highly diverse metabolic status" - maybe clarify the meaning here.

The references are appropriate.

---

## Author Comment (AC1) · 15 Mar 2018

**Response to referee 1 M. Aranguren-Gassis**

*General comments*
*In the manuscript "Dynamics of phytoplankton and heterotrophic bacterioplankton in the western tropical South Pacific Ocean along a gradient of diversity and activity of diazotrophs", Van Wambeke et al. present estimations of heterotrophic prokaryotic production in the Western Tropical South Pacific region, and they explore the causes of its variability, focusing on autotrophic activity and nutrient availability. The data presented in the paper are very valuable, as the carbon budget in the oligotrophic regions is still a topic of debate, in part because of lack of data to adequately characterize the bacterial contribution to it. The data presented are a great mix of observations and experiments, and the analysis made have great potential. However, the way the paper is written makes the information confusing and the conclusions vague. The title doesn't reflect the contents of the paper, the methods are not complete, the discussion is not well structured or clear, and the conclusions are not in line with the results highlighted in the discussion (See details in specific comments). The English and the writing needs profound review.*

We thank Dr. Maria Aranguren-Gassis for her constructive comments. We respond to her below using regular black fonts and provide references to modified text in our manuscript using regular blue fonts. A native English speaker will check the revised version of the ms.

*Specific comments*
*- Title: The authors mention in the title a gradient of diversity and activity of diazotrophs, but such gradient is not shown in the paper. They base part of their discussion in different groups of diazotrophs described during the same cruise by other authors (lines 530-541), but as they described it, it is not a diversity gradient, just differences in the dominant genera. The diazotrophs activity gradient is not clear either in the paper. I would suggest for the title to focus more on the analysis made to elucidate the factors controlling the bacterioplankton activity in different regions of an oligotrophic system.*

The title was modified as:
'Dynamics and controls of heterotrophic prokaryotic production in the western tropical South Pacific Ocean: links with diazotrophic and photosynthetic activities.'

*- Abstract: The abstract is a good summary of the paper, but I think some parts can be removed:*
*\* Line 28: the i.e. can be removed, it makes the sentence too long, and it is not necessary*
It is done

*\* Line 30: The BGE estimation doesn't provide useful information here*
We removed this sentence

*\* Lines 33-36: I don't find this information about the bloom developed along the paper, I think this should be removed from here.*
We agree that the study of a bloom collapse at site LDB did not constitute the main focus of the paper. This part was rephrased and moved up in the abstract.

*- Methods: Some variables have a lot of weight in the discussion but methods are not described. For example Nitrogen fixation rates, community respiration and GPP, or nutrients*

*(nitrate, nitrite, and phosphate) concentrations. At least a brief description of the methods should be included, even if they have been explained somewhere else.*

Since the submission of our manuscript, papers devoted to the study of nitrogen fixation rates (Bonnet et al., 2018) and nutrients and organic matter distribution (Moutin et al., 2018) have been published in the OUTPACE special issue, in which detailed methodologies are available. However, to provide guidance to the reader, we added a few sentences describing methodology in M&M section 2.1 as follows:

'Besides measurements of chlorophyll *a,* BP, PP, $T_{DIP}$ and DOC described below, other data presented in this paper include hydrographic properties, nutrients, $N_2$fix, for which detailed protocols of analysis and considerations for methodology are available in Moutin et al. (2017; 2018) and Bonnet et al. (2018). Briefly, DIP and nitrate concentrations were measured using standard colorimetric procedures on a AA3 AutoAnalyzer (Seal-Analytical). The quantification limits were 0.05 µM for both nutrients. $N_2$ fixation rates were measured using the $^{15}N_2$ tracer method in 4.5 L polycarbonate bottles inoculated with 5 ml of $^{15}N_2$ gas (99 atom % $^{15}N$, Eurisotop). Note that the risk of underestimation by this bubble method was checked by subsampling and fixing 12 ml of each bottle after incubation and analyzing the dissolved $^{15}N_2$ with a Membrane Inlet Mass Spectrometer.'

For $O_2$ based metabolic rates, since the paper by Lefevre et al. (this issue) is still not submitted, and as Dark Community Respiration are used to estimate bacterial growth efficiency, the protocol has been briefly developed in M&M section 2.5 as follows:

'Rates of dark community respiration (DCR) were used to estimate bacterial growth efficiency (see discussion). Briefly, DCR was estimated from changes in the dissolved oxygen ($O_2$) concentration during dark incubations of unfiltered seawater (24 h) carried out at LD stations, *in situ* on the same mooring lines used for PP$_{in\ situ}$ (Lefevre et al., this issue). Quadruplicate Biological Oxygen Demand bottles were incubated in the dark at each sampled depth. The concentration of oxygen was determined by Winkler titration. DCR was calculated as the difference between initial and final $O_2$ concentrations, and the mean standard error of volumetric DCR rates was 0.28 µmol $O_2$ dm$^{-3}$ d$^{-1}$.'

In addition, data on integrated DCR rates were added in Table 2.

*- Lines 109-112: The criteria used for stations selections is not stated. Even if it is described in other papers, a better explanation should be included here because it can affect the interpretation of the results.*

The end of Results 3.1 section was modified as follows:

'The transition between the MA and WGY areas is particularly evidenced by an enhanced degree of oligotrophy in the WGY area. WGY area was characterized by dcm depths deeper than 115 m (Table 2), deep nitracline (130 m) and nitrite peaks around 150 m and detectable amounts of phosphate at the surface (> 100 nM, Moutin et al., 2018). A detailed analysis of the vertical distribution of nutrients and organic matter made it possible to identify two groups of stations within the MA area, each having common biogeochemical characteristics: one group between 160 and 170°E called WMA for 'Western Melanesian Archipelago' clustered SD1, 2, 3 and LDA and a second group South of Fidji called EMA for 'Eastern Melanesian Archipelago' clustered SD6, 7, 9 and 10 (Moutin et al., 2018). Main biogeochemical differences between these two groups of stations were related to shallower depths for phosphacline (20m), nitracline (76m), dcm (82 m), in WMA group (see Table 2 and Figure 5 b, c in Moutin et al., 2018). The EMA group had intermediate depths for these parameters in comparison to WMA and WGY, (phosphacline 44 m, nitracline 100 m and dcm 105 m).

Although geographically included within the MA area, LDB corresponded to a particular bloom condition and is therefore presented and discussed separately.'

*- Line 113: It should be mention here that the sampling in LD stations was Lagrangian.*
This has been done

*- Lines 151 and 161: The word "occasionally" is too vague, please specify at least how many times.*
This has been done (9 times for time kinetics, 5 times for concentration kinetics)

*- Lines 165-179: The incubation time used should be specified.*
This has been done as follows:
'Incubations times lasted 4 (western stations) to 24h (south Pacific Gyre area) and were chosen according to expected $T_{DIP}$.'

*- Lines 185 and 207: Are the cast numbers necessary? I think they can be removed.*
Yes, they have been removed

*- Line 230: The authors talk about a gradient, but they don't specify what kind of gradient. A gradient of productivity? A gradient of diazotrophs activity?*
To improve clarity, we changed the first sentence of paragraph 3.1 as follows:
'The longitudinal transect started North West of New Caledonia, crossed the Vanuatu and Fidji Arcs and finished inside the western part of the ultra-oligotrophic South Pacific Gyre.'

*- Lines 240-245: Authors say that nutrients and organic matter distribution allowed them to distinguish two regions, but those data are not shown in the paper. At least, a description of the differences between the regions should be included. For example, you can include some extra data in Table 2 with nutrients and organic matter concentration, or whichever criteria used to identify those two regions.*
See previous response describing the modification of the end of section 3.1

*- Lines 236-246 are confusing and need to be rewritten.*
See previous response describing the modification of the end of section 3.1

*- Line 249: "Averaged per SD station, the dcm fluctuated..." I don't understand what that means and what the following ranges refer to.*
We think that the misunderstanding comes from our definition of dcm: in the ms introduction, we defined 'dcm' as 'the depth of the deep chlorophyll maximum' (see line 48 of the first version) instead of the more common use of this acronym as the 'deep chlorophyll maximum'. In the revised version, we changed that definition as follows:
'The South Pacific gyre (GY) is ultra-oligotrophic, and is characterized by deep UV penetration, by deep chlorophyll maximum (dcm) depth down to 200 m, and by a 0.1 µM nitrate ($NO_3$) isocline near 160 m (Claustre et al., 2008b; Halm et al., 2012)'
When necessary, we added the word 'depth' after dcm, to help the reader understand that the descriptions of LD site variability with time refer to the vertical change of the dcm depth.

*- Line 269: I suppose what authors meant is that TDIP increased with depth, not the vertical profiles themselves.*
Yes of course, this has been corrected.

*- Line 281: I don't think the periodicity of the dcm fluctuation is evident in figure 5. That fluctuation and the increase of fluorescence in the afternoon are not well described. Those patterns are not apparent in the figures, and not statistical analysis is presented. However, I think those results can be avoided because authors don't mention them along the discussion at any point.*

We recognize that such patterns are much more visible when plotting fluorescence versus density instead of depth. See for example what it would give for site LDA:

[Figure]

We agree partly with the referee's recommendations. The part focusing on the increase of fluorescence in the afternoon which is not developed in the discussion has been deleted. However, the part describing internal waves, particularly on site LDA, is important because such temporal variability makes difficult to compare profiles of biogeochemical parameters or biological fluxes when they were not made at the same time. Thus, this part was kept.

*-Line 286: Values from In situ and on deck incubation cannot be directly compared, as temperatures for the incubations are different. Particularly for samples from depths below the mixer layer. So authors shouldn't highlight a higher value from on deck incubations without a proper analysis of correspondence between on deck and in situ estimations.*

A new paragraph was added at the beginning of section 3.4 as follows:
'There are several limitations with comparing $PP_{deck}$ and $PP_{in\ situ}$. Incubation on mooring lines for 24h dawn-to-dawn is considered to be a good compromise by JGOFS recommendations (JGOFS, 1988), as temperature and light are close to *in situ* conditions (except UV). Incubation on deck, under simulated *in situ* conditions suffers from biases related to the use of artificial screens to mimic light attenuation with depth, and also from biases related to temperature decrease for deeper samples, as they are incubated at sea-surface temperature. During our cruise, at each LD site on day 5, we used both incubation methods, and did not

sample the same CTD cast: $PP_{in\ situ}$ was sampled at 3:00 AM while $PP_{deck}$ was sampled at 9:00 AM. At site LDA, differences between the mean $IPP_{in\ situ}$ and $IPP_{deck}$ were particularly high. Besides artifacts related to light and temperature described above, one of the explanation could be due partly to internal waves (de Verneil et al., 2017; Bouruet-Aubertot et al., this issue) as for instance the dcm depth changed from 69 m to 87 m between the 3:00 ctd cast and the 9:00 ctd cast at the site LDA on day 5. At the site LDB, the bloom collapsed rapidly and a trend with time was clearly detected, making the comparison between both methods impossible, even with only a time lag of 6h. For this reason, and to keep relative comparisons consistent, we used only $PP_{deck}$ data when exploring relationships between BP, PP, $N_2$fix and $T_{DIP.}$'

de Verneil, A., Rousselet, L., Doglioli, A. M., Petrenko, A. A., Maes, C., Bouruet-Aubertot, P., and Moutin, T.: OUTPACE long duration stations: physical variability, context of biogeochemical sampling, and evaluation of sampling strategy, Biogeosciences Discuss., https://doi.org/10.5194/bg-2017-455, in review, 2017

Bouruet-Aubertot, P., Cuypers, Y., Le Goff, H., Rougier, G., Picheral, M., Doglioli, A., Yohia, C., de Verneil, A., C ffin, M., Petrenko A., Lefevre, D., Moutin, T. Longitudinal contrast in Turbulence along a 19S section in the Pacific and its consequences on biogeochemical fluxes. Biogeosciences Discuss., this issue, in prep.

We checked that both types of rates were not mixed in any of our interpretations: Time variability on sites LDA, LDB and LDC only used $PP_{in\ situ}$ (Figs 5, 6, 7), whereas analysis of longitudinal trends, and correlations with BP only included $PP_{deck}$ data (Figs 2, 3, 8 and 10). Because in the first version of the manuscript, the correlations described on Fig 8 included both types of rates, we now have done the analysis including only $PP_{deck}$ data. This is why values cited on equation linking BP with PP have been slightly changed in the revised version (see below), but the conclusions did not change.

*-Lines 324-326: Figure 8 shows a linear regression analysis, but in the text, authors present the correlation coefficient (r), but not in the figure or in the text they mention the significance of the fitting. With the correlation coefficient, authors shouldn't interpret the results as a dependency, because the correlation between two data sets doesn't indicate causality of one of the variables from the other.*

We added p in the equations:

log BP=0.842 log PP - 0.57, n=47, r = 0.26, p=0.04 and
log BP=0.808 log PP - 0.53, n=90, r = 0.67, p < 0.001

for samples where $T_{DIP}$ was $\leq$ 100 h and > 100 h, respectively

log BP=0.752 log $N_2$fix - 0.78, n=39, r = 0.52, p < 0.001 and
log BP=0.438 log $N_2$fix - 0.31, n=55, r = 0.43, p < 0.001

for samples where $T_{DIP}$ was $\leq$ 100 h and > 100 h, respectively

*- Line 331-334: Those N2fix temporal trends are not shown anywhere. If authors don't want to show them in a figure, use some statistics to state those patterns.*

To better illustrate the longitudinal variability, we modified Fig. 3b and plotted integrated N$_2$fix rates, as well as the N$_2$fix rates to bacterial nitrogen demand ratio instead of the bacterial nitrogen demand. The new Fig. 3 can be found at the end of the responses to the reviewer's comments.

N$_2$fix temporal trend and the concomitant changes in the ratio of N$_2$fix to bacterial nitrogen demand at site LDB was explicitly detailed and we added statistic results for comparison of this ratio between site LDA and site LDC.

*- Line 393-404: The definition of the regions is not well described. Some suggestions:*
*\* Use the same terms to name them along the entire manuscript.*
In the revised version of our manuscript, we paid attention to use the terms WMA (Western Melanesian Archipelago), EMA (eastern Melanesian Archipelago) and WGY (western part of the South Pacific Gyre) when they were cited all along the ms.

*\* Show them on the map (figure 1)*
We modified Figure 1 that can be seen at the end of the responses to the reviewer's comments.

*\* Refer here to Table 2, and complete the table with the criteria described here (nitracline depth).*
Distinction between WMA, EMA and WGY group of stations is based, among other parameters, on dcm depth which is already included in Table 2. Biogeochemical characteristics explaining differences between WMA, EMA and WGY stations are based on differences between depths of phosphacline, nitracline and dcm, which more or less followed the same trend, i.e. intermediary depths for EMA stations (between WMA stations, shallower, and WGY stations, deeper). Depth of nitracline and phosphacline are indicated in Table 2 of Moutin et al (2018) and depth of phosphacline was added for WMA and EMA in Figure 4. Hence, we think it is unnecessary to add nitracline and phosphacline depths in Table 2.

*\* Define what "distinct nutrient distributions" means or show it in a plot*
See previous response describing the modification of the end of section 3.1. Sentences describing dcm, nitracline and phosphacline depths are not anymore used in section 4.1

*\* In figure 4 put in different colors or patterns the profiles for each region*

The new version of Fig. 4 is presented at the end of the responses to the reviewer's comments.

*- Line 422: "after filtration and removal... producers" corresponds to methods section.*

This part of the sentence was removed

*- Line 431-432: Why is the information in the last sentence of this paragraph relevant? Please, elaborate*

The last sentence was moved upper in this paragraph in a more appropriate place. So that the paragraph is now organized as follows:
'Bacterial growth efficiencies (BGE) obtained from biodegradation experiments ranged 6–12 %, with a small labile fraction of DOC (only 2–5 % of biodegradable DOC in 10 days). Thus, the bulk DOC was mainly refractory, although DOC concentration was higher in surface waters (Moutin et al., 2018). Large stocks of DOC, with C/N ratio ranging 16 to 23 have also

been reported in the surface waters of the SPG (Raimbault et al., 2008). Both high C/N ratios and a small labile fraction suggests that this surface bulk pool of DOC is probably largely recalcitrant due to UV photodegradation or photooxidation (Keil and Kirchman, 1994; Tranvik and Stephan, 1998; Carlson and Hansel, 2015) or by action of the microbial carbon pump (Jiao et al., 2010). Small BGE and small labile fraction could also be due to strong resource dependence as low nutrient concentrations cause low primary production rates, and low transfer across food webs. Indeed, Letscher et al. (2015) also observed surface DOC recalcitrant to remineralization in the oligotrophic part of the eastern tropical south Pacific. But as shown by these authors, incubation with microbial communities from the twilight zone, provided by addition of an inoculum concentrated in a small volume, allowed DOC remineralization. This was explained by relief from micronutrient limitation or potential role for co-metabolism of relatively labile DOC provided by the inoculum with more recalcitrant DOC. Our enrichment experiments effectively suggest nutrient limitation, although the second hypothesis could not be excluded.

In order to better explain the variability of BGE measurements, we also estimated this parameter indirectly, using dark community respiration (DCR) and BP data that were measured simultaneously. We converted DCR to carbon units assuming a respiratory quotient $RQ = 0.9$, and computed BGE from Ze-integrated BP and DCR assuming either bacterial respiration (BR) to be within a range of 30 % of DCR (BGE=BP/(BP+DCR*0.9*30 %), Rivkin and Legendre, 2001; del Giorgio and Duarte, 2002) or 80 % of DCR (BGE=BP/(BP+DCR*0.9*80 %), Lemée et al., 2002; Aranguren-Gassis et al., 2012). The range of these indirect estimates of BGE were similar to those obtained from the biodegradation experiments: 3–12 % at site LDA, 4–17 % at site LDB and 2–7 % at site LDC. Note, however, an increasing trend from day 1 to day 5 at site LDB: on average 8 % on day 1, 10 % on day 3 and 12 % on day 5. Including all direct and indirect estimates, the mean (± sd) BGE was $8 \pm 4$ % (n = 21).

*- Lines 445-462: This paragraph is not well linked to the rest of the discussion. There is not any mention of the present paper results, and it is not clear the contribution of the present paper to the debate described in the paragraph. Please, elaborate.*

We completely re-organized section 4.2: We now start by the paragraph discussing metabolic balance. Then, the paragraph introducing PP and GPP was modified as follows:
'It is known that the in vitro [14]C method measures an intermediate state between net PP and GPP. However, Moutin et al. (1999) showed that GPP could be reasonably estimated from daily net PP determined from dusk-to-dusk as: GPP = 1.72 * PP, a ratio also used by others authors (Loisel et al., 2011). On the other hand, dealing with the assumptions made to convert hourly leucine incorporation rates to daily BCD, there are many biases that have been largely debated, including mostly those resulting from daily variability, assumptions on BGE or BR (Alonzo-Saez et al., 2007; Aranguren-Gassis et al., 2012b), carbon to leucine conversion factors (Alonso-Saez et al., 2010), and light conditions of incubations including UV (Ruiz-Gonzales et al., 2013). For this cruise, we measured data to discuss BGE variability. Daily variability is also taken into account using results from previous experiments in the South Pacific Gyre (BIOSOPE cruise, Van Wambeke et al., 2008). Finally, we also discuss one largely unexplored bias, related to the ability of *Prochlorococcus* to assimilate leucine in the dark.'

Loisel, H., Vantrepotte, V., Norkvist, K., Mériaux, X., Kheireddine, M., Ras, J., Pujo-Pay, M., Combet, Y., Leblanc, K., Dall'Olmo, G., Mauriac, R., Dessailly, D., and Moutin, T. : Characterization of the bio-optical anomaly and diurnal variability of particulate matter, as

seen from scattering and backscattering coefficients, in ultra-oligotrophic eddies of the Mediterranean Sea, Biogeosciences, 8, 3295-3317, doi:10.5194/bg-8-3295-2011, 2011

Then, we followed by 3 paragraphs describing successively BGE variability, daily variability and finally the correction factors linked to the assimilation of leucine by *Prochlorococcus*. The short paragraph on daily variability as requested by the second referee:
'Bias introduced when converting hourly to daily BP rates was not studied here, but we use a dataset obtained in the South Pacific Gyre (Van Wambeke et al., 2008) to estimate conversion errors. During the BIOSOPE cruise, vertical profiles of BP were acquired using the leucine technique along the euphotic zone, every 3 h up to 72 h, at three selected sites using Lagrangian sampling strategy. For the 3 series of profiles, standard deviations of IBP with time were 13 % (n = 13), 16% (n = 16) and 19 % (n = 9). Thus, standard errors represented 3.6, 4.2 and 6.1 % of the mean BPI, respectively. We used the average value of this percentage (5 %) to estimate the bias introduced by the conversion from hourly to daily IBP estimates of the OUPACE cruise. '

*- Lines 482-488: This paragraph is confusing and needs rewriting*

The paragraph was corrected as follows:
'Using flow cytometry cell sorting of samples labelled with $^3$H-leucine during the OUTPACE cruise, Duhamel et al. (in revision) demonstrated the mixotrophic capacity of *Prochloroccoccus*, as this phytoplankton group was able to incorporate leucine, even under dark conditions, albeit at lower rates than under light conditions. This group was found to be able to assimilate ATP, leucine, methionine as well as glucose, a single C-containing molecule (Duhamel et al., in revision, and ref therein). To date, few organic molecules have been tested and mainly those including N, P or S sources. As leucine assimilation by *Prochloroccoccus* was significantly detected in dark incubations in all examined samples, it will affect BP measurements. We thus corrected (BP$_{corr}$) to represent the assimilation of leucine in the dark by heterotrophic bacteria alone. Based on Duhamel et al. (in revision), leucine assimilation by HNA+LNA bacteria in the dark corresponded on average (± sd) to 76 ± 21 % (n = 5, range 44–100 %) of the activity determined for the community including *Prochlorococcus* (HNA + LNA + Proc).'

*- Lines 490-492: I don't understand the calculations described in here*

We changed Fig. 10, presenting BCD$_{corr}$ instead of BP$_{corr,}$ so that GPP and bacterial carbon demand could be directly compared.

*- Lines 490-500: What is the overall contribution of all these calculations to the paper?*

The originality is that we compiled the influence of cumulated biases affecting GPP, BCD and their ratio by considering the propagation of errors related to the variability of the reproducibility of measurements, daily variability, BGE variability, and the *Prochlorococcus* assimilation of leucine.

*- Lines 506-509: I don't see the relation between the three first lines of the paragraph and the following ones.*

The first sentence has been moved later in this paragraph when we talk about N and C limitation in WGY area. The second sentence has been removed.

*- Lines 548-549: I think it is incorrect to deduce competition ability from this correlation*

We removed this term. This paragraph has been modified as follows:
We found that the slope of the regression between $N_2$fix rate and BP was greater and the correlation was better within the mixed layers and when the $T_{DIP}$ is low (<100 h), i.e. in areas characterized by low phosphate availability (Moutin et al., 2008), whereas in this waters variability of PP explained slightly the vaiability of BP (r=0.26). A better correlation between BP and $N_2$fix than between BP and PP, would suggest that bacteria may have been more dependent on the availability of a new N source than a new C source, which is in agreement with results from enrichments at LDA and LDB. Because $T_{DIP}$ was lower in areas of high $N_2$fix rates, it is likely that DIP drawdown was due to diazotrophs, which while bringing new sources of N, reduced DIP availability. Indeed, at the site LDB within the mixed layers, BP increased after N addition alone but also after P addition alone, which suggests a direct limitation of BP by N and potentially a cascade effect of P addition towards heterotrophic prokaryotes : P would directly stimulate $N_2$ fixers which rapidly would transfer new N and labile C available to stimulate BP'

*- Conclusions: The conclusions don't reflect the discussion or even the results presented*

From line 584, the end of the conclusion was modified as follows:
'Our results provide a unique set of simultaneous measurements of BP, PP and $N_2$fix rates in the WTSP. BP obtained in the WTSP was in the same range as those previously measured in the GY area eastern of 140°W. BGE was low and the bulk DOC was found to be mainly refractory. In surface, nitrogen and relatively DIP depleted waters, BP was more strongly correlated to $N_2$fix than to PP, while the more traditional coupling of BP with PP occurred deeper in the euphotic zone. This suggests that in the surface layers with greater diazotrophic activity, BP was more dependent on the availability of new N from $N_2$ fixers than on the availability of fresh C from primary producers, which was also demonstrated through enrichment experiments. We showed that the interpretation of PP and BP fluxes based on instantaneous methods (radioisotopic labelling) needs regular tests to verify the major methodological biases and conversion factors hypotheses. In particular, to make conclusions about the metabolic state of oceanic regions, it is necessary to consider the variability of all conversion factors used to estimate carbon-based GPP and BCD. In addition, the use of the leucine technique to estimate BP should be used with caution in N-limited environments due to the potential mixotrophy by cyanobacteria.'

*- References: There are six not published references in the list, and some of them have data with a lot of weight in the discussion.*

This is inevitably a problem during the editing process of special issues. Among the 6 publications not published in December 2017, three of them are now in Biogeosciences Discussions and one has been published in Applied Microbial Ecology:

Bonnet, S., Caffin M., Berthelot H., Grosso, O., Benavides, M., Helias-Nuninge, H., Guieu, C., Stenegren, M. and Foster, R.: In depth characterization of diazotroph activity across the Western Tropical South Pacific hot spot of $N_2$ fixation, Biogeosciences Discuss., doi.org/10.5194/bg-2017-567, 2018

Dupouy, C., Frouin, R., Tedetti, M , Maillard, M.., Rodier, M., Lombard, F., Guidi, L., Picheral, M., Duhamel, S., Charrière, B., and Sempéré, R.: diazotrophic *Trichodesmium* influences ocean color and pigment composition in the South West tropical Pacific, Biogeosciences Discuss., doi.org/10.5194/bg-2017-570, in review, 2018

Moutin, T., Wagener, T., Caffin, M., Fumenia, A., Gimenez, A., Baklouti, M., Bouruet-Aubertot, P., Pujo-Pay, M., Leblanc, K., Lefevre, M., Helias Nunige, S., Leblond, N., Grosso, O. and de Verneil, A.: Nutrient availability and the ultimate control of the biological carbon pump in the Western Tropical South Pacific Ocean. Biogeosciences Discuss., /doi.org/10.5194/bg-2017-565, 2018.

Tenorio, M., Dupouy C., Rodier, M., and Neveux, J. Filamentous cyanobacteria and picoplankton in the South Western Tropical Pacific Ocean (Loyalty Channel, Melanesian Archipelago) during an El Nino episode, Appl. Microb. Ecol.*,* doi.org/10.3354/ame01873, 2018

*- Table 1: Better put the PP units on the table. It would be really helpful to group the rows by region, so it is easier to follow the description in Lines 403-415.*
These 2 recommendations have been followed.

*- Figure 1: a more general map to locate the cruise area would be useful. In the legend, the fourth line will be easier to read using "respectively".*
The legend has been modified as suggested. We changed Figure 1 and Nouméa and Papeete are now indicated to help for localization, letters and numbers of stations are more contrasted, and we added colored squares to identify the groups of stations corresponding to WMA, EMA and WGY areas. The new Fig 1 is presented at the end of this author comment

*- Figure 2: I suggest using the same units for the two panels to make them directly comparable. Put stations number in both plots.*
We added station numbers in both plots. We found unnecessary to use the same units as the goal of such plots is just to illustrate trends. Scientists working with bacterial production data are more familiar with hourly units, closer to what has been really measured. Comparison is however possible on Figure 8a (volumetric rates), 2 and 10 (integrated rates) where the same units are used for PP (GPP) and BP (BCD) rates.

*- Figure 3: * The units on the panel B are incorrect, it is mmol.*
This has been corrected
*\* Why are data at station 13 missing? I don't find the explanation.*
The explanation was on the legend of Table 2. We added the sentence also on Figure 3.
*\* As I understand, the bars on figure 3 represents the values of each variable in every station, so these plots are not histograms, they are bar charts.*
Yes, the Legend has been modified
*\* Bars should be represented by the corresponding error bars.*
We added error bars (standard errors)

*-Figure 4: this figure needs to be improved. Some suggestions:*
*\* Make the station names consistent with other figures.*
This has been done
*\* Group the profiles by region, with color or pattern, or make 3 panels, one for each region.*

*\* Talking about this figure in the text (line 269) authors use the phosphacline. Represent the phosphacline here to help.*

Fig. 4 has been modified and now includes 4 panels with plots separating EMA, WMA and EGY stations. The depth of phosphacline was added. The new Fig. 4 is presented at the end of the responses to the reviewer's comment.

*\* I would incorporate the profiles for the Long stations in figure 9, as you use the information in the discussion of the experiments.*

$T_{DIP}$ vertical profiles at sites LDA, LDB and LDC were done only once per LD station, on day 5 and not on day 2 when other parameters presented in Figure 9 were measured. As we explained in the ms, due to the high internal waves at site LDA, as well as the bloom collapse at site LDB, there was too much variability between day 2 and day 5 at LD stations and therefore, we preferred not to insert $T_{DIP}$ data on Figure 2.

*- Figures 5, 6 and 7:*
*\* Vertical axis in the second panel should say fluorescence instead of chl, as you explain in the text (e.g., line 280)*

This has been corrected

*\* When describing these figures in the text, you use the density. Including the density level lines in the figures will be a good idea.*

Yes it is. We modified the Figure accordingly. For example the new Fig. 5 is presented at the end of the responses to the reviewer's comment.

*\* Explain in the legend what the vertical lines in the second panel mean, and make them the same color than the profiles in the third panel.*

We added a sentence in the legend:
'On the in vivo fluorescence graph, vertical bars show the 12:00 AM ctd cast sampled for BP each day (1 to 5) with corresponding colours used for plotting BP vertical profiles'.
In fact, the color code was already done. We now use thicker lines to better see the differences.

*- Figure 10: What is it that you represent in here is not clear for me, not in the text or the figure legend. Please make the description of the calculation more clear and explain better the meaning and the interpretation. Use station names consistent with the rest of the paper.*

This has been modified, in the text as well as in the legend. We also modified the figure, plotting bacterial carbon demand instead of BP. A copy of the new Fig. 10 can be found at the end of the responses to the reviewer's comments.

*Technical corrections*
 *– Line 23: space is missing between With and N2 loom*
*– Lines 94 and 97, the period is missing*
*- Lines 110 and, the first n in Lagrangian is missing*
*- Line 119: I think authors meant experiments (in plural)*
*- Line 234: a bracket is missing before the references*

All five corrections done

*- Lines 236-246 are confusing and need to be rewritten.*
See above the modification of the end of section 3.1

*- Line 249: "Averaged per SD station, the dcm fluctuated..." I don't understand what that means and what the following ranges refer to.*
See response above (dcm versus dcm depth)

*- Line 269: Vertical profiles cannot increase or decrease with depth, I assume you are talking here about TDIP decreasing with depth. Please rewrite.*
Yes sorry, this has been corrected

*- Line 278: Please, check this sentence. Space is missing between down and in. "Comumn" I suppose means columns with l. There is a comma instead of a period after the bracket. The sentence doesn't make sense in general.*
The sentence was modified as follows:
'Site LDA presented variable dcm depth over time (63 to 101 m, Table 2), as illustrated by patches of in vivo fluorescence moving up and down the water column with time, along a band of 40 m height (Fig. 5). However, the dcm depth corresponded to a stable density horizon (σt 23.55 ± 0.04 kg m$^{-3}$), and thus this fluctuation in dcm depth corresponded to internal waves characterized by a periodicity of about 2 per day (Fig. 5).'

*- Line 287: delete the bracket before IPPDECK*
Done

*- Line 327: PP instead of BP - Line 345: Those abundances, I guess are bacterial abundances*
This has been corrected. We now only use BP data and DOC data in biodegradation experiments

*- Line 346: a decimal point is missing in 014*
*- Line 347: the lowest instead of the lowerest instead of the lower*
The 2 corrections were done

**New versions some Figures.**

[Figure]

**Figure 1** Stations locations during the OUTPACE cruise. The white line shows the ship track (data from the hull-mounted ADCP positioning system). In dark green WMA (western Melanesian Archipelago) included SD1, 2, 3 and LDA; in light green, EMA: eastern Melanesian Archipelago included SD 6, 7, 9 and 10 and in blue WGY (western gyre) included stations SD13, 14, 15 and LDC.
Figure courtesy of T Wagener.

[Figure]

**Figure 2** Distribution of primary production (a) and heterotrophic prokaryotic production (b) along the OUTPACE cruise transect. Interpolation between sampling points in contour plots was made with the Ocean Data View software (VG gridding algorithm, Schlitzer, 2004). The white dots in (a) correspond to the average ± sd of the dcm depth at each station. The white rectangles mask abnormal extrapolation due to the absence of PP data.

[Figure]

**Figure 3** a) Distribution of integrated heterotrophic prokaryotic production (IBP) and primary production (IPP$_{deck}$) along the transect, data were integrated over the euphotic zone. b) Distribution of integrated N$_2$ fixation rates and of ratio N$_2$ fixation rates to bacterial nitrogen demand (I N2fix/I-BND, assuming a bacterial C/N ratio of 5 and no nitrogen excretion) along the transect. Data were integrated down to the deepest sampled depth for N$_2$ fixation rates. Data plotted for sites LDA, LDB and LDC correspond to BP, PP$_{deck}$ and N$_2$fix measured on day 5. Error bars are standard errors (s.e.) derived from triplicate measurements at each depth (BP, PP$_{deck}$, N$_2$fix rates). For BP, error bars also take into account the daily variability, and final s.e. were calculated after propagation of errors. PP obtained at SD13 was abnormally low (55 mg C m$^{-2}$ d$^{-1}$) and was excluded; BP and N$_2$fix rates were not measured at this station. .

[Figure]

**Figure 4** Vertical distributions of phosphate turnover times ($T_{DIP}$) in groups of stations WMA (a), EMA (b), WGY (c) and other stations (d). At the long-duration stations LDA, LDB and LDC, $T_{DIP}$ profiles were determined at day 5 (bold lines). Horizontal bar in a (WMA) and b (EMA) delineates the mean phosphacline depth (mean ± sd: 20 ± 7 m, and 44 ± 10 m, respectively) as determined by Moutin et al. (2018). At WGY (c), DIP concentrations were > 100 nM at all depths.

[Figure]

**Figure 5** Evolution of surface PAR, in vivo fluorescence, PP and BP at the site LDA. Time units in local time, day1 was February 26, 2015. BP samples were taken at the 12:00 PM ctd cast, while samples for PP$_{in\ situ}$ were taken at the 3:00 AM ctd casts (day 1, 3 and 5). On the in vivo fluorescence graph, vertical bars show the 12:00 AM ctd cast sampled for BP each day (1 to 5) with corresponding colours used for plotting BP vertical profiles.

[Figure]

**Figure 10** Distribution of integrated bacterial carbon demand corrected for *Prochlorococcus* assimilation) and based on a 8% BGE (I-BCD$_{corr,}$ grey bars), gross primary production derived from IPP$_{deck}$ (I-GPP, blue bars) along the transect. Error bars are standard errors, calculated using propagation of errors.* indicates stations where the hypothesis Ho that the 95% confidence interval of the difference (I-BCD minus I-GPP) includes zero was rejected.

---

## Author Comment (AC2) · 15 Mar 2018

**Response to Anonymous Referee #2**

*In this article the authors present the results from a study into bacterial and primary production in the tropical south Pacific ocean. The paper fits perfectly within the scope of Biogeosciences. I found the article interesting to read with some very interesting insights into the carbon balance of this part of the Tropical South Pacific, a region that has been rather less studied than some of the other oceanic provinces.*
*While the actual methods used can be considered as relatively classic in the domain, their application to this little studied area is novel. Indeed, although several authors have worked in the Tropical South Pacific, the vast majority of these studies have looked at either N2 fixation alone or have been conducted in the coastal areas near to Islands.*
*This data from the open ocean is particularly interesting and novel. The assumptions of the methods are appropriate and are clearly outlines.*

*I am wondering why was the ratio 400ml of bacterial 'inoculum' chosen for addition to 2.6L?*
It was a typo error that we have corrected. We added 400 ml in a volume of 1.6 L so that the dilution factor was 20%. In oligotrophic environments, adding only 10% usually leads to a very long lag phase.

*The conclusions are appropriate and provide some interesting insights into what is limiting bacterial production in this part of the ocean. Notably, it appears that available N is the limiting factor - which of course underlines the importance of N2 fixing organisms in this environment, as has been already shown by other work from this group.*

*I was a little perplexed as to why some results were shown in the methods section Pg 4, line 135.*
In the M&M section, we wanted to justify why we used in vivo fluorescence only to describe shape of vertical profiles, depths of dcm, high frequency time evolution, whereas discrete measures of chlorophyll by fluorometry were used to estimate and compare chlorophyll biomass stocks. This is why we prefer to keep this information in the M&M. However, it does not need to be developed to a great extent, and the last paragraph was reduced as follows:
'Due to the heterogeneity at the time of sampling and the nature of the populations present, i.e. essentially different fluorescence yields over depth and species (Neveux et al., 2010), the overall correlation of in vivo fluorescence (chl iv) with chl a was very patchy (chl a=1.582 * chl iv + 0.0241, n = 169, r = 0.61). Thus in vivo fluorescence was used to track high frequency variability at the LD sites, the shape of vertical profile's distributions and the location of the dcm, as well as longitudinal trends. But fluorometric discrete data (chl a) was always used when calculating and comparing integrated stocks.'

*The results section is sufficient to support the conclusions - I have one comment here though - it was a little awkward to have quite a few associated datasets were in other articles - it was a bit difficult to do a "stand-alone" review. But the authors do clearly give credit for other work and they clearly indicate what their new additions are.*
This is inevitably a problem during the process of special issues. Among the 6 publications not published in December 2017, 3 of them are now submitted in Biogeosciences Discussions and one in press has been published:
Bonnet, S., Caffin M., Berthelot H., Grosso, O., Benavides, M., Helias-Nuninge, H., Guieu, C., Stenegren, M. and Foster, R.: In depth characterization of diazotroph activity across the

Western Tropical South Pacific hot spot of $N_2$ fixation, Biogeosciences Discuss., doi.org/10.5194/bg-2017-567, 2018

Dupouy, C., Frouin, R., Tedetti, M , Maillard, M.., Rodier, M., Lombard, F., Guidi, L., Picheral, M., Duhamel, S., Charrière, B., and Sempéré, R.: diazotrophic *Trichodesmium* influences ocean color and pigment composition in the South West tropical Pacific, Biogeosciences Discuss., doi.org/10.5194/bg-2017-570, in review, 2018

Moutin, T., Wagener, T., Caffin, M., Fumenia, A., Gimenez, A., Baklouti, M., Bouruet-Aubertot, P., Pujo-Pay, M., Leblanc, K., Lefevre, M., Helias Nunige, S., Leblond, N., Grosso, O. and de Verneil, A.: Nutrient availability and the ultimate control of the biological carbon pump in the Western Tropical South Pacific Ocean. Biogeosciences Discuss., /doi.org/10.5194/bg-2017-565, 2018.

Tenorio, M., Dupouy C., Rodier, M., and Neveux, J. Filamentous cyanobacteria and picoplankton in the South Western Tropical Pacific Ocean (Loyalty Channel, Melanesian Archipelago) during an El Nino episode, Appl. Microb. Ecol.*,* doi.org/10.3354/ame01873, 2018

*The experiments and calculations are well described and will allow for replication by other scientists.*

*The Title clearly reflects the contents, particularly if we take into account the whole group of papers from the Outpace experiment.*

*The abstract is clear but I wondering if the last sentence should not appear earlier in the text, it does seem to be a little be disconnected from the rest of the text. Perhaps the authors can rephrase it if they wish to leave it as a last sentence or move it up.*

This sentence was moved upwards in the abstract

*Yes, the article is well structured, clear and I really enjoyed reading it. The language is fluent and clear and the appropriate formulae and correction factors used are presented clearly when needed.*

*Concerning the tables: Table 1 and 2 : both of these tables are a little blurry – maybe check that in a revised version?*

It is because we had to paste an image in an A4 format whereas the initial table was in a Landscape format, not accepted in the edited version. We will work with the editor to make sure that our tables appear clearly in the final version of the manuscript. Dealing with the content of Table 1, we also organized referenced areas in a more logical order, roughly from west to east.

*Also, can the authors add the units into the table (I know they are in the legend, but I always find it easier to follow when they are in the table itself).*

It is done

*Table 5 : can the authors add in if its the mean +/- the SD or the SE?*

It is ± SE. It was added on the Table 5

*Figure 1 : is a little hard to see - but maybe it's my printout - nevertheless, can the authors check that the figure is clear and not blurry.*

In the original ppt version, figures are not blurry. In addition, we made a new version of figure 1 where letters are better contrasted and the different groups of stations (WMA, EMA and WGY) are indicated. See the new Figure 1 at this end of the responses to the reviewer's comments

*Figure 3 : check the format of the legend titles (add in uppercase letters when needed).*
This has been corrected

*Figure 4: why did the authors choose to put in the black dotted lines? It rather draws the eye at the cost of the other profiles.*
We made a new Fig.4 (please see at the end of the responses to the reviewer's comments)

*Overall, can the authors unify the format of the axis titles on the figures - some have () some do not. Also can they check the clarity of the contour maps and the colour of the words/numbers on the graphics - sometimes they are hard to read (see figs. 5-7b).*
Parenthesis for units have been added everywhere. In their original version in ppt format, the Figures are not blurry We will take care of that during editing process of the revised version

*pg 9 line 341 : non signficant for PP*
This has been corrected

*line 250 : what do the authors mean here 'determined by fluorometry'? Don't both methods employ fluorometrey (Turner vs CTD)?*
Yes, you are right. The term 'determined by fluorometry' was removed

*pg 11, line 420 : 6-12% is not that low.*
We removed the term 'low'

*line 440 : check spelling of Lemee here (it's okin the Refs).*
This has been corrected

*Paragraph starting 455: negative NCP values have also been observed in the oligotrophic water off-shore of New Caledonia (Pringault et al. Biogeosciences 2007).*
Yes, this reference was added in the ms.

*I agree with the authors that calculating up hourly incubation values to daily ones is fraught with errors. Do the authors have an estimate of how much error may be introduced from these factors?*
Diel variability of BP was studied in the eastern south Pacific gyre at three sites: in the open sea away from Marquesas Islands, in the center of the South Pacific gyre, and in the eastern part of the south Pacific gyre (Van Wambeke et al., 2008). At these sites, a Lagrangian sampling strategy was used and we followed BP every 3 h up to 72 h long. From this study, the coefficients of variation (SD/Mean ratio) of euphotic zone-integrated hourly BP were 13, 16 and 19%. With the number of profiles varying between 9 and 16 per site, the standard error (SE represented on average 5 % of the mean. We used this value in the context of the OUTPACE cruise to estimate the SE introduced by the conversion from hourly to daily bacterial carbon demand, using propagation of errors (SE. related to triplicate variability of BP, and SE related to daily variability). The corresponding SE is now plotted in Fig. 3a. We also added this information in the text.

*It is interesting to note that Prochlorococcus could be responsible for up to 56% of leucine uptake - this could have some very strong implications for BCD calculations and hence, ecosystem metabolism calculations in areas where Prochlorococcus is abundant.*
*What about the diazotrophs? Do they take up leucine? Is there any information on this?*

Yes, co-author S. Duhamel did some cell sorting of *Crocosphera*-like cells which peaked at 60 m at the site LDC. She detected significant uptake of leucine by these cells in light and dark conditions. Although the activity was significantly detected per cell, due to their low abundance, their participation to the bulk uptake of leucine was very low. These results will be presented in a manuscript in preparation outside of this Special Issue. Moreover, assimilation of dissolved organic nitrogen labeled with $^{15}$N and observed by Nanosims technique also suggested assimilation by *Trichodesmium*, although the hypothesis that this labeling could be obtained indirectly after a transfer via epibiontic heterotrophic bacteria could not be ruled out (Benavides et al., 2017).

*530 : what is an artificial diazotroph culture?*
We wanted to insist on the fact that the information did not concern natural environment. Of course a culture is not e natural environment. We removed the term 'artificial'

*570: not sure what the authors mean here in the sentence starting "They also showed..." - can the authors revised this?*
The sentence was modified as:
'They also showed a highly dynamic *Crocosphaer*a growth and decay during diel cycles survey, suggesting rapid switch between cell growth and mortality processes, such as grazing and viral infection.'

589: what do the authors mean by "highly diverse metabolic status" - maybe clarify the meaning here.
This last sentence was removed

**New versions of some Figures**

[Figure]

**Figure 1** Stations locations during the OUTPACE cruise. The white line shows the ship track (data from the hull-mounted ADCP positioning system). In dark green WMA (western Melanesian Archipelago) included SD1, 2, 3 and LDA; in light green, EMA: eastern Melanesian Archipelago included SD 6, 7, 9 and 10 and in blue WGY (western gyre) included stations SD13, 14, 15 and LDC. Figure courtesy of T Wagener.

[Figure]

**Figure 4** Vertical distributions of phosphate turnover times ($T_{DIP}$) in groups of stations WMA (a), EMA (b), WGY (c) and other stations (d). At the long-duration stations LDA, LDB and LDC, $T_{DIP}$ profiles were determined at day 5 (bold lines). Horizontal bar in a (WMA) and b (EMA) delineates the mean phosphacline depth (mean ± SD: 20 ± 7 m, and 44 ± 10 m, respectively) as determined by Moutin et al. (2018). At WGY (c), DIP concentrations were > 100 nM at all depths.

---

## Author Comment (AC3) · 20 Mar 2018

We apologize for that, but an error occurred when presenting the new Figure 4 in our author comments AC1 and AC2. The codes of stations were correct, the color codes were correct, and consequently 4a (blue) is WGY and not WMA, 4b (dark green) is WMA, and not EMA, and 4c (light green) is EMA, but not WGY. The mean phosphacline depths of EMA and WMA group of stations have been moved accordingly.

Note also that we were informed by an e-mail dated on March 17, 2018 that the ms by Duhamel et al 'Mixotrophic metabolism by natural communities of unicellular cyanobacteria in the western tropical South Pacific Ocean' has been accepted for publication in 'Environmental Microbiology'. It seems to us that this information is relevant as a part of the discussion on our ms discuss the bias related to assimilation of leucine by *Prochlorococcus*.

[Figure]

**Figure 4** Vertical distributions of phosphate turnover times ($T_{DIP}$) in groups of stations WGY (a), WMA (b), EMA (c) and other stations (d). At the long-duration stations LDA, LDB and LDC, $T_{DIP}$ profiles were determined at day 5 (bold lines). Horizontal bar in b (WMA) and c (EMA) delineates the mean phosphacline depth (mean ± SD: 20 ± 7 m, and 44 ± 10 m, respectively) as determined by Moutin et al. (2018). At WGY (a), DIP concentrations were > 100 nM at all depths.